# Predictive Differential Training Guided by Training Dynamics

**Fanqi Wang[1]\*, Weisheng Tang[1]\*†, Landon Harris[1], Hairong Qi[1]†, Dan Wilson[1], Igor Mezić[2]**
[1]University of Tennessee, Knoxville   [2]University of California, Santa Barbara
{fwang20,lharri73}@vols.utk.edu,
{wtang7,hqi,dwilso81}@utk.edu,  mezic@ucsb.edu

## Abstract

This paper centers around a novel concept proposed recently by researchers from the control community where the training process of a deep neural network can be considered a nonlinear dynamical system acting upon the high-dimensional weight space. Koopman operator theory (KOT), a data-driven dynamical system analysis framework, can then be deployed to discover the otherwise non-intuitive training dynamics. Taking advantage of the predictive power of KOT, the time-consuming Stochastic Gradient Descent (SGD) iterations can be then bypassed by directly predicting network weights a few epochs later. This "predictive training" framework, however, often suffers from gradient explosion especially for more extensive and complex models. In this paper, we incorporate the idea of "differential learning" into the predictive training framework and propose the so-called "predictive differential training" (PDT) for accelerated learning even for complex network structures. The key contribution is the design of an effective masking strategy based on a dynamic consistency analysis, which selects only those predicted weights whose local training dynamics align with the global dynamics. We refer to these predicted weights as high-fidelity predictions. PDT also includes the design of an acceleration scheduler to adjust the prediction interval and rectify deviations from off-predictions. We demonstrate that PDT can be seamlessly integrated as a plug-in with a diverse array of existing optimizers (SGD, Adam, RMSprop, LAMB, etc.). The experimental results show consistent performance improvement across different network architectures and various datasets, in terms of faster convergence and reduced training time (10-40%) to achieve the baseline's best loss, while maintaining (if not improving) final model accuracy. As the idiom goes, a rising tide lifts all boats; in our context, a subset of high-fidelity predicted weights can accelerate the training of the entire network!

## 1 Introduction

The advent of cutting-edge hardware (Li et al., 2014) and the development of parallel processing techniques (Li et al., 2020) have greatly accelerated the training process of Deep Neural Network (DNN). However, enhancing the fundamental techniques of DNN training continues to be a significant challenge. From the inception of Stochastic Gradient Descent (SGD) (Robbins & Monro, 1951), which has since become a mainstay in DNN training, numerous techniques have been proposed to increase the efficiency of the underlying optimization task, including, for example, learning rate annealing and momentum (Sutskever et al., 2013), RMSprop (Tieleman & Hinton, 2012), and Adam (Kingma & Ba, 2014). In addition to these first-order optimizers, second-order alternatives (Martens, 2010) utilizing curvature information or second-order derivatives of the loss function have been explored to potentially enable more efficient convergence. Despite these advancements, gradient-based methods are still inherently iterative, requiring repeated gradient computations and weight adjustments throughout the network. This iterative burden manifests a **fundamental limitation** of SGD and its variants, which lies at the core of the computationally expensive training process.

---

\*Equal contribution.
†Correspondence to Weisheng Tang (wtang7@utk.edu) and Hairong Qi (hqi@utk.edu).

The concept of *differential learning*—where different parts of the network can exhibit different learning behaviors during training—has emerged as a promising direction to address this limitation. Differential learning can be layer-specific (Devlin et al., 2019; He et al., 2019) or parameter-specific (Tieleman & Hinton, 2012; Duchi et al., 2011a), allowing for more targeted optimization. The Adam optimizer (Kingma & Ba, 2014), for instance, adaptively computes individual learning rates for different parameters. While differential learning takes adaptive approaches on how parameters are updated, it does not fundamentally address the limitation of the iterative optimization process itself.

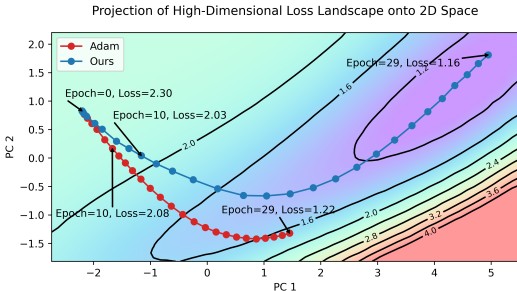

Figure 1: Comparison of training trajectories and loss landscapes between Adam and the proposed PDT. (Trained on CIFAR-10 using AlexNet.)

Recently, a novel interpretation of the DNN training process has been proposed, mainly by researchers from the control community (Redman et al., 2022; Dogra & Redman, 2020; Manojlovic et al., 2020; Tano et al., 2020; Redman et al., 2024) – If it is intuitive to consider a pre-trained DNN as an inherently nonlinear static system acting upon the inputs, then *the DNN "training process" itself is a "nonlinear" dynamical system acting upon the high-dimensional "weight space"*! It is a discrete dynamical system since the weights of a DNN evolve over each iteration (or epoch) according to the optimization process adopted. This drastically different interpretation has led to the establishment of a novel mathematical framework for learning. Koopman Operator Theory (KOT) (Mezić, 2005), a powerful data-driven dynamical system analysis tool, is often adopted to exploit the underlying dynamics in the seemingly non-intuitive training process of a DNN. Taking advantage of the predictive power of KOT, the time-consuming SGD iterations can be bypassed by directly predicting network weights a few epochs later (Dogra, 2020; Dogra & Redman, 2020; Tano et al., 2020). We refer to these approaches as *predictive training*.

However, practical challenges quickly emerge. The absence of actual gradient descent means that convergence cannot be guaranteed, and the framework is sensitive to disturbances in the weight space, leading to error accumulation across iterations. As the network scales, the previous Koopman-based predictive training framework becomes increasingly ineffective. This issue is mostly due to the lack of adaptive mechanisms when applying prediction-based acceleration. That is, existing predictive training approaches tend to accept *all* predicted weights without checking if the prediction is of "high-fidelity" or not. This often leads to gradient explosion, especially for more extensive and complex models.

The **key observation** is that even though KOT is a powerful predictive tool for studying traditional small-scale control problems, when dealing with DNN whose parameter dimension reach into the millions or even billions, the quality of prediction tends to be highly inhomogeneous across the entire weight space. Hence, the predictive learning has to be "selective" – only high-fidelity predictions should be selected to effectively accelerate learning.

In this paper, we propose *predictive differential training* (PDT), where acceleration by prediction is selectively applied based on a dynamic consistency analysis. This principled approach identifies parameters that are in a stable, predictable phase of their evolution by ensuring their local dynamics align with the global system dynamics modeled from the training history. This selective acceleration is conceptually similar to various adaptive learning rate methods. For instance, Adagrad (Duchi et al., 2011b) targets acceleration at rare features; Momentum (Rumelhart et al., 1986) prioritizes weights with the largest recent velocity; and the popular Adam optimizer (Kingma & Ba, 2014) employs a combined strategy. Fig. 1 illustrates the compelling effectiveness of PDT over Adam through a visual

comparison of the training trajectory and loss landscape. The contributions of the proposed PDT can be summarized as follows:

- We propose the Predictive Differential Training (PDT) framework that selectively applies predictive updates to effectively accelerate training.

- We design a dynamic consistency analysis as a masking strategy to conduct prediction. It selects parameters whose local training dynamics align with the global dynamics, allowing PDT to identify parameters that are in a stable, predictable phase of their evolution.

- We demonstrate that PDT can be seamlessly integrated as a plug-in with numerous existing optimizers, such as SGD, Adam, RMSprop, Shampoo, and LAMB, while maintaining computational efficiency through epoch-level predictions.

- We validate PDT's effectiveness across diverse network architectures (from FCN to ViT-Huge), datasets (from CIFAR-10 to ImageNet), and learning paradigms (from supervised to self-supervised), demonstrating its scalability and robustness under various conditions. [1]

## 2 BACKGROUND AND RELATED WORK

The key notion of Koopman analysis is the representation of a (possibly nonlinear) dynamical system as a linear operator on a typically infinite-dimensional space of functions (Mezić, 2021; 2005; Mezić & Banaszuk, 2004; Wang et al., 2024). Koopman-based approaches directly contrast with standard linearization techniques that consider dynamics around a nominal solution. Indeed, Koopman analysis can yield linear operators that accurately capture fundamentally nonlinear dynamics.

**Koopman Operator Theory.** As a brief description, consider a discrete-time dynamical system $\mathbf{x}_{i+1} = T(\mathbf{x}_i)$, where $\mathbf{x}_i \in \mathbb{R}^n$ is the current state and $\mathbf{x}_{i+1}$ is the next state after applying the potentially nonlinear mapping $T$. Consider also a vector-valued observable $\mathbf{g}(\mathbf{x}) \in \mathbb{R}^m$. The evolution of observables under this mapping can be described as

$$\mathbf{g}(\mathbf{x}_{i+1}) = \mathbf{g}(T(\mathbf{x}_i)) = \mathcal{K}\mathbf{g}(\mathbf{x}_i). \tag{1}$$

where $\mathcal{K}$ operates on the vector space of observables and maps $\mathbf{g}(\mathbf{x}_i)$ to $\mathbf{g}(\mathbf{x}_{i+1})$. $\mathcal{K}$ is referred to as the "Koopman operator" that is associated with the fully nonlinear dynamical system. The Koopman operator is linear, but also infinite-dimensional. As such, for dynamical systems with a pure point spectrum for observables (Mezić, 2020), its action can be decomposed according to

$$\mathbf{g}(\mathbf{x}_{i+1}) = \mathcal{K}\mathbf{g}(\mathbf{x}_i) = \sum_{k=1}^{\infty} \lambda_k \phi_k(\mathbf{x}_i)\mathbf{c}_k, \tag{2}$$

where $\lambda_k$ is an eigenvalue associated with the eigenfunction $\phi_k(\mathbf{x})$, which can be evaluated at either the initial state $\mathbf{x}_0$ or any intermediate state $\mathbf{x}_i$. $\mathbf{c}_k$ is the reconstruction coefficient, also known as the "Koopman mode", which represents the projection of the observable function $\mathbf{g}$ onto the eigenspace. It immediately follows that $\mathbf{g}(\mathbf{x}_{i+\tau}) = \sum_{k=1}^{\infty} \lambda_k^{\tau} \phi_k(\mathbf{x}_i)\mathbf{c}_k$ for any $\tau \in \mathbb{N}$. This has provided a convenient and general framework to "predict and control" a given dynamical system. Each Koopman mode evolves over time with its frequency and decay rate governed by the imaginary and real components, respectively.

Koopman-based techniques are particularly useful in a data-driven setting because they only require measurements of observables. As such, they can be implemented even when the underlying model dynamics are unknown.

**Dynamic Mode Decomposition (DMD).** When using Koopman-based approaches, it is critical to identify a suitable *finite* basis for representing the infinite-dimensional Koopman operator. Dynamic Mode Decomposition (DMD) (Schmid, 2010) is one standard approach for inferring Koopman-based models. It uses least-squares fitting techniques to approximate a finite-dimensional linear matrix operator, $A$, that advances high-dimensional measurements of a system forward in time:

$$\mathbf{g}(\mathbf{x}_{i+1}) \approx A\mathbf{g}(\mathbf{x}_i) \tag{3}$$

where $A$ is an approximation of the Koopman operator, $\mathcal{K}$, in Eq. 1, restricted to a measurement subspace spanned by direct measurements of the state $\mathbf{x}$. Since the weight space of a neural network is

---

[1] Our source code is publicly available at `https://github.com/aicip/PDT`.

a *fully observable* system, we define $\mathbf{g}(.)$ to be the identity function in this work. That is, $\mathbf{w}_i = \mathbf{g}(\mathbf{w}_i)$. In practice, we often use "snapshots" of the system arranged into two data matrices, $W_i$ and $W_{i+1}$, where columns of these matrices indicate measurements (i.e., network weights) taken at a certain time, and $W_{i+1}$ is $W_i$ shifted by one time step. Hence,

$$W_{i+1} \approx AW_i, \tag{4}$$

and $A$ can be solved by $A = W_{i+1}W_i^{\dagger} = W_{i+1}V\Sigma^{-1}U^T$, where $W_i = U\Sigma V^T$ is the Singular Value Decomposition (SVD), and $W_i^{\dagger}$ denotes the pseudo-inverse of $W_i$. A comprehensive discussion of DMD and its variants has been provided in Kutz et al. (2016).

**DNN Training as a Dynamical System.** There have been a few works in recent years that adopt Koopman-based approaches to accelerate the training process of a general-purpose DNN model (Dogra & Redman, 2020; Tano et al., 2020; Manojlovic et al., 2020). (Dietrich et al., 2020) is generally considered the first work that establishes the connection between KOT and acceleration of numerical computation. Dogra (2020) is also one of the pioneer works but with a focus specifically on neural networks for solving differential equations. Generally speaking, these works take advantage of the predictive power of the KOT framework to directly predict network weights a few epochs later, thus bypassing the time-consuming SGD iterations. However, we show in Fig. 2 that these methods tend to fail for larger network structures as the network size increases.

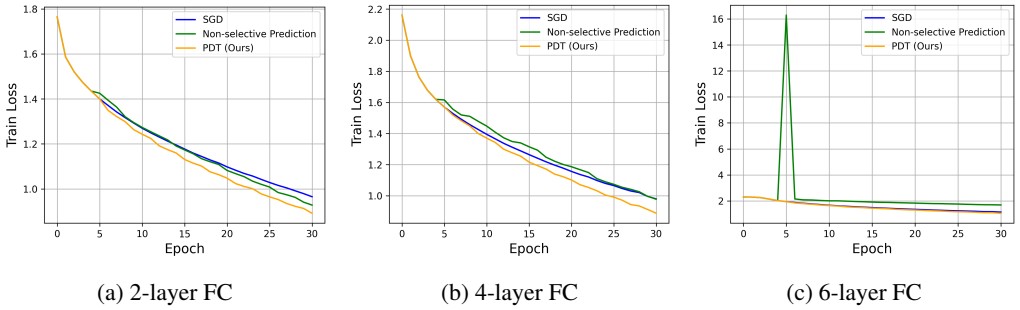

|       (a) 2-layer FC       |       (b) 4-layer FC       |       (c) 6-layer FC       |

Figure 2: Performance comparison on CIFAR-10 using fully connected (FC) networks with varying depths, among SGD (iterative), PDT (predictive-differential), and the non-selective prediction, i.e., Koopman-based predictive training where the predicted weights are applied to *all* parameters without checking the prediction quality (Tano et al., 2020). Batch size=256, lr=0.01. In our setup, for every three epochs of SGD, predictions are performed for the next five steps.

Directly predicting the evolution of neural network weights, by bypassing SGD, is inherently difficult due to the complex and unstable nature of training dynamics. The loss landscape is highly non-convex, filled with local minima, saddle points, and flat regions (Goodfellow et al., 2014), while the effective dynamics is non-stationary (Ghorbani et al., 2019), as both gradients and curvature shift as training progresses (Sagun et al., 2017). In addition, neural systems can exhibit chaotic or highly sensitive regimes, where small perturbations quickly amplify and destabilize predictions (Li & Ravela, 2021; Engelken et al., 2023). This challenge is compounded by the stochastic noise introduced through mini-batch sampling.

Small prediction errors are highly sensitive and cumulative, risking divergence in the absence of continual gradient correction (Andrychowicz et al., 2016). Moreover, predictors trained in one context often fail to generalize across architectures and datasets, highlighting the difficulty of extracting universally valid patterns (Wichrowska et al., 2017; Metz et al., 2019). Together, these factors make weight prediction a fundamentally unstable and error-prone task.

Beyond these general challenges, a number of prior works have attempted to predict future weights directly, such as Introspection (Sinha et al., 2017), WNN (Jang et al., 2023), and the more recent NiNo (Knyazev et al., 2024). These methods typically rely on a separately learned predictor—either element-wise regression models or graph-based networks—trained on curated checkpoint datasets before being applied to a new target model. The effectiveness therefore depends on the predictor's meta-training distribution, and the inference cost grows with model size because predictions are applied at per-weight or per-edge granularity. In contrast to these learned predictors, the proposed

PDT adapts to the heterogeneous training dynamics of different parameters, enabling it to sustain network growth and provide a viable solution to the weight-prediction challenge. As a result, PDT functions as a lightweight plug-in without requiring external checkpoint datasets or per-weight inference overhead. The efficiency of PDT has been validated on several benchmark models (e.g., AlexNet, ResNet, and ViT), datasets (e.g., CIFAR-10 and ImageNet), spanning both supervised and self-supervised tasks.

## 3 METHODS

Let us first use a toy example to demonstrate the effect of accelerating the learning of a *subset* of variables to motivate the concept of differential learning. Consider the function,

$$f(x, y, z, u, v, w) = x^2 + y^2 + \sin(z) + u^2 - \cos(v) + w^2 + xy + y\sin(z) + uvw,$$

which involves six variables: $x, y, z, u, v, w$. To find the minimum of this function, we employ a simple GD optimization with a learning rate of $0.01$. GD takes 53 steps to reach a loss value below our predefined threshold (0.1).

We then explore an alternative optimization strategy where the variables $x, y, z$ use a learning rate three times faster than that of the standard process, while $u, v, w$ are optimized at the normal rate but employing the updated values of $x, y, z$. See Fig. 8 in Appendix A.1 for the acceleration trajectory, where the trajectory maintains the same direction for $x$ and $y$ but reaches the threshold in just 25 steps. This example demonstrates by strategically identifying a subset of variables and simply increasing their learning rate, the training can be accelerated by about 53%. We also apply the proposed PDT to the same optimization problem and it reaches the threshold in 27 steps. See Fig. 9 in Appendix A.1. This toy example demonstrates the principle behind the idiom, a rising tide lifts all boats!

### 3.1 PDT TRAINING FRAMEWORK

The PDT Training Framework addresses three key questions: 1) when to enable prediction, 2) how to integrate predictions with existing optimizers, and 3) which parameters should undergo accelerated updates. The complete PDT workflow and the mechanism involved in a single acceleration step are illustrated in Fig. 3. The "prediction" block (`Pred`) is automatically but strategically placed among the baseline optimization blocks (`OPT`), acting as a plug-in enhancement within the existing optimization framework. Training begins with a "Burn-in stage," where the model is trained using the baseline optimizer for several epochs to accumulate a sufficient history of weight snapshots. Following this stage, a prediction step is performed with an adaptive interval, $\tau$, to achieve accelerated learning.

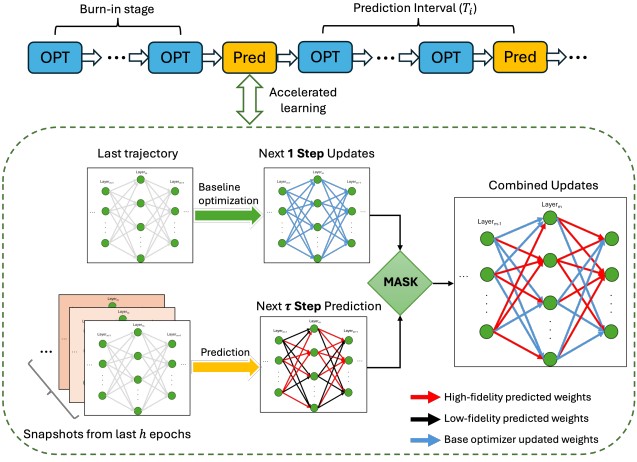

Figure 3: Illustration of the proposed PDT framework and the detailed mechanism for a $\tau$-step prediction, where qualified (or high-fidelity) predicted weights (red) and standard SGD-derived weights (blue) are integrated that accelerate the training of the entire network.

The bottom part of Fig. 3 provides a detailed illustration of how qualified predicted weights and standard SGD-derived weights are integrated together to achieve accelerated learning, as showcased in the toy example. The mask is governed by the dynamic consistency analysis to be elaborated in Sec. 3.2. If no element in the mask satisfies the criteria, then standard SGD-based optimization takes over. The pseudocode of the complete PDT algorithm is presented in Appendix A.2.2.

The amount of computation required to perform a DMD-based prediction is comparable to that of a GD operation. It is important to note that the prediction operations are much less frequent (once for several epochs) compared to the standard GD operations (multiple times per epoch, depending on the batch size). Considering that PDT requires fewer epochs to reach convergence (see Table 1), it can lead to significant computational savings and efficiency enhancements in the training of large-scale neural networks. The theoretical time and space complexity analysis is provided in Appendix A.3, and a detailed profiling and overhead analysis is provided in Appendix A.8.

## 3.2 DYNAMIC CONSISTENCY ANALYSIS

The prediction step begins by applying DMD to the weight snapshots, $W_i$ and $W_{i+1}$, which yields a finite-dimensional approximation of the Koopman operator, $A$, as in Eq. 4. Since $A$ is of high dimension, $N \times N$, where $N$ is the number of weights (or parameters) of the network, it is computationally intractable to solve directly. Hence, we resort to the *Standard DMD* algorithm (Tu et al., 2014). This method projects the dynamics onto a low-rank subspace to efficiently compute the eigenvalues $\Lambda$ and the high-dimensional DMD modes $\Phi$ without computing $A$ directly (see Appendix A.2.1 for the derivation). To practically implement the spectral prediction from Eq. 2, the predicted weight vector is computed from Eq. 5:

$$\mathbf{w}_{i+\tau}^{\text{pred}} = \Phi \Lambda^\tau \Phi^\dagger \mathbf{w}_i \tag{5}$$

where $\Phi$ is the matrix whose columns are the DMD modes (approximating the eigenfunctions $\phi_k$), $\Lambda$ is the diagonal matrix containing the corresponding eigenvalues $\lambda_k$, and $\Phi^\dagger$ denotes the Moore-Penrose pseudoinverse. The term $\Phi^\dagger \mathbf{w}(i)$ projects the current state onto the DMD modes, calculating the Koopman mode amplitudes $\mathbf{c}_k$ in Eq. 2.

Our approach is based on the principle that DMD extracts the dominant patterns of the entire system's dynamics. However, at any given training stage, different parameters may exhibit varying degrees of alignment with these global patterns. Parameters experiencing rapid transitions, or local instabilities, may not conform to the global linear dynamics assumption underlying DMD. By leveraging the spectral components ($\Phi$, $\Lambda$) derived from the low-rank approximation of $A$, we can perform a multi-step prediction through a more stable spectral evolution process using Eq. 5, which provides a prediction for the system's global dynamics and also offers a perspective on the prediction for each parameter. The challenge, however, is how to determine whether such a prediction for each parameter has "high-fidelity" or "low-fidelity".

In fact, the correlation between the quality of prediction and training effectiveness has been heavily studied. From a neuroscience perspective, the quality of predictions made by neurons is intricately linked to their learning dynamics (Schultz et al., 1997; Friston, 2010). Accurate predictions lead to more stable and efficient learning, while poor predictions need stronger synaptic adjustments to improve future performance.

Therefore, we design a masking mechanism to identify parameters whose current local dynamics align with the system's global dynamics, based on the following two principles.

**The acceleration effectiveness criterion.** The absolute weight change between the predicted weight, $\mathbf{w}_{i+\tau}^{\text{pred}}$, and the current weight, $\mathbf{w}_i^{\text{opt}}$, at each epoch, $i$, should be *larger* than the absolute weight change from the one-step optimization, $\mathbf{w}_{i+1}^{\text{opt}} - \mathbf{w}_i^{\text{opt}}$, to enable accelerated learning; simultaneously, we impose an upper bound of $\tau$ multiples of $\mathbf{w}_{i+1}^{\text{opt}} - \mathbf{w}_i^{\text{opt}}$, where $\tau$ is the prediction step, to ensure stable convergence. That is,

$$\|\mathbf{w}_{i+1}^{\text{opt}} - \mathbf{w}_i^{\text{opt}}\| < \|\mathbf{w}_{i+\tau}^{\text{pred}} - \mathbf{w}_i^{\text{opt}}\| \leq \tau \|\mathbf{w}_{i+1}^{\text{opt}} - \mathbf{w}_i^{\text{opt}}\|, \tag{6}$$

This criterion ensures that the prediction provides a significant advancement beyond what single-step optimization would achieve, making the acceleration worthwhile. See Appendix A.4 for convergence guarantee analysis.

**The dynamic consistency criterion.** The direction of weight change from prediction should align with the local gradient-based dynamics. That is, the temporal evolution captured by the global DMD analysis should be consistent with the current local optimization trajectory. Specifically:

$$sign(\mathbf{w}_{i+k,j}^{\text{pred}} - \mathbf{w}_{i,j}^{\text{opt}}) = sign(\mathbf{w}_{i+1,j}^{\text{opt}} - \mathbf{w}_{i,j}^{\text{opt}}), \tag{7}$$

where $j$ is the index for each element in the weight vector and $k = \{1, \cdots, \tau\}$. Note that when $k = 1$, $\mathbf{w}_{i,j}^{\text{pred}} = \mathbf{w}_{i,j}^{\text{opt}}$. This criterion ensures that each step of the prediction interval follows the same trend of growth as that of the local optimization.

Based on these two principles, a mask, $\mathbf{m}$ can be constructed with its element equal to 1 if both Eqs. 6 and 7 are satisfied; otherwise, the corresponding element is zero. This dynamic consistency analysis evaluates these two criteria independently for each parameter. Parameters satisfying both criteria are deemed to be in a predictable evolutionary phase, allowing safe application of temporal acceleration through the global dynamic model. Parameters failing these criteria may be experiencing complex local behaviors (such as rapid transitions, oscillations, or instabilities) that deviate from the global linear dynamics assumption, requiring fallback to gradient-based updates. Note that Eq. 7 is a rigid criterion to enforce not only that the final predicted weight changes align with the local optimization direction, but also that every intermediate step in the predicted trajectory maintains direction consistency.

## 4 EXPERIMENTS

### 4.1 GENERALIZATION STUDY OF PDT

We evaluate the effectiveness of the proposed PDT framework in accelerating learning across a variety of popular neural network architectures with different scales, including Fully-Convolutional-Network (FCN) (3.9M parameters), AlexNet (57M parameters), ResNet-50 (25.6M parameters), ViT-Base (86.4M parameters), and ViT-Huge (632M parameters). PDT shows a significant advancement over previous prediction-only Koopman-based methods limited to simpler models [e.g., (Tano et al., 2020) with 2.9M trainable parameters]. We also use a range of optimizers, including SGD, SGD with momentum, and AdamW (Loshchilov & Hutter, 2019). Our validation spans multiple benchmark datasets, from CIFAR-10 to the more challenging ImageNet-1K.

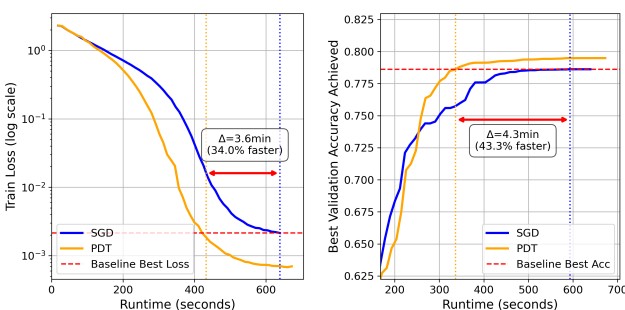

Figure 4: Analysis of PDT's Optimization and Generalization Efficiency. Trained on CIFAR-10 using AlexNet, batch size=256, lr=0.05, with CosineAnnealingLR scheduler.

Fig. 4 illustrates the efficiency of PDT by showing the training curve on CIFAR-10 using AlexNet. We assess PDT's performance from two perspectives: (1) Optimization Efficiency, measured by the **Time to Baseline Best Train Loss** (or **TTB-Loss**), quantifies the speed at which PDT converges on the training objective. (2) Generalization Efficiency, measured by the **Time to Baseline Best Validation Accuracy** (or **TTB-Acc**), reflects the speed at which PDT obtains a useful, well-generalized model. Note that all runtime metrics reported in this paper represent the total wall-clock time, fully inclusive of all computational overheads introduced by PDT, such as SVD decomposition, multi-step prediction, and mask generation. The detailed training curves on various network structures can be found in Fig. 10 in Appendix A.5. A detailed profiling of these overheads is provided in Appendix A.8.

In all experiments, unless otherwise specified, we use the past five epochs to form the snapshot with a one-epoch interval to predict weights in the next five steps. Prediction starts from the 5th epoch.

A comprehensive study of PDT's performance under different training configurations (batch sizes, learning rates, and optimizers) and hyperparameters [i.e., prediction steps ($\tau$), prediction interval ($T_i$), starting epoch ($T_0$), and past snapshot counts ($h$)] can be found in Appendix A.6, demonstrating robust performance across various training hyperparameters.

As elaborated in Sec. 3, the computational load of the Koopman-related calculations is comparable to that of batch-level updates. However, since we apply these calculations at the epoch level, the overhead introduced by DMD is effectively compensated by the acceleration in loss reduction. We observe from both Table 1 and Fig. 10 that the proposed PDT consistently achieves the best training loss of the Baseline but in fewer number of epochs without sacrificing performance. All experiments were repeated with five random seeds to ensure reliability. Unless otherwise specified, all results in the tables are reported as mean ± standard deviation over five runs.

Table 1: Runtime comparison. FCN and AlexNet are trained on a single Nvidia RTX A6000 GPU, while ResNet-50, ViT-Base, and ViT-Huge are trained on three Nvidia H100 (80 GB) GPUs. Using the same experimental setup and hyperparameter configurations as in Fig. 10.

| Model | Baseline Optimizer | TTB-Loss (s) | | TTB-Acc (s) | | Runtime Reduction (%) | |
|---|---|---|---|---|---|---|---|
| | | Baseline | PDT | Baseline | PDT | Train Loss | Val. Acc. |
| FCN | SGD | 2174.32 | 1313.52 | 2088.58 | 1424.14 | **39.59** | **31.81** |
| AlexNet | SGD | 683.93 | 430.91 | 531.30 | 347.11 | **37.00** | **34.67** |
| ResNet-50 | SGD-M | 110063.72 | 88752.33 | 121449.60 | 92133.34 | **19.36** | **24.14** |
| ViT-Base | AdamW | 259241.21 | 232810.62 | 296028.36 | 243097.58 | **10.20** | **17.88** |
| ViT-Huge | AdamW | 725564.86 | 653854.05 | 741220.54 | 660711.80 | **9.88** | **10.86** |

The last column in Fig. 10 illustrates a so-called "masked ratio curve" unique to PDT, where it tracks the percentage of predictions accepted according to the masking strategy described in Sec. 3.2. We make two interesting observations. First, we observe that the masked ratio always starts with higher values in the early stage of the training process, then generally decreases as training progresses. This trend aligns naturally with how GD-based baselines behave. That is, in the early stage of the training process, the loss landscape is typically easier to optimize, leading to faster reduction in loss and more stable gradient directions, which in turn allows more weights to pass the masking criteria, hence a higher masking ratio. Later in training, as the GD-based optimizer approaches (local) minima, gradients tend to oscillate more around the optimum, making it more challenging to predict, hence less percentage of predicted weights being accepted. *More interestingly*, we observe that smaller networks on simpler tasks (FCN/AlexNet on CIFAR-10) show a relatively more gradual reduction in the masked ratio, while larger networks on more complex tasks (ResNet-50/ViT on ImageNet) exhibit a much sharper reduction of masked ratio, especially at the early stage of the training process. This pattern implies that for large networks on large datasets, the training dynamics are inherently complex and challenging to predict. The higher masked ratio at the initial training stage is primarily attributed to the steep loss landscape and large gradients. As the model converges and gradients diminish, the intrinsic complexity of the training dynamics becomes dominant, resulting in a rapid reduction in the percentage of weights that can be convincingly predicted (according to the proposed masking strategy). A detailed analysis of how the mask distribution evolves across different layers of the network during training is provided in Appendix A.7. A study on the impact of different initial learning rates on the masked ratio evolution is presented in Appendix A.10.

To further demonstrate the versatility of PDT as a plug-in enhancement, we extended our evaluation to include a broader range of optimizers [SGD, SGD with momentum, Adam, AdamW, RMSprop, Shampoo (Gupta et al., 2018), and LAMB (You et al., 2020)] while keeping the network architecture and other configurations fixed. The results in Table 2 show that PDT consistently reduces the time to reach baseline best loss across these optimizers.

We also extend our evaluation to the domain of self-supervised learning (SSL). We select Sim-Siam (Chen & He, 2021), a prominent non-contrastive method, as our testbed. SimSiam's training dynamics, which are driven by a stop-gradient mechanism and a negative cosine similarity objective, are fundamentally different from those of supervised learning. The results, summarized in Table 3, demonstrate that PDT's advantages generalize effectively to SSL.

Table 2: Impact of baseline optimizers on PDT performance. Trained on CIFAR-10 using AlexNet, batch size=256, momentum=0.9, with CosineAnnealingLR scheduler.

| Optimizer | lr | Final Accuracy | Best Train Loss | TTB-Loss (s) | Runtime Reduction (%) |
|---|---|---|---|---|---|
| SGD | 0.1 | $0.7930 \pm 0.0023$ | $0.0002 \pm 0.0000$ | $665.27 \pm 9.08$ | **19.67** |
| PDT | | $0.7978 \pm 0.0032$ | $0.0002 \pm 0.0000$ | $534.41 \pm 12.64$ | |
| Momentum | 0.001 | $0.6672 \pm 0.0068$ | $0.8609 \pm 0.0166$ | $752.74 \pm 9.62$ | **41.06** |
| PDT | | $0.7298 \pm 0.0051$ | $0.5358 \pm 0.0165$ | $443.68 \pm 8.75$ | |
| Adam | 0.0005 | $0.7952 \pm 0.0063$ | $0.0001 \pm 0.0000$ | $779.13 \pm 11.81$ | **14.87** |
| PDT | | $0.8050 \pm 0.0050$ | $0.0002 \pm 0.0000$ | $663.28 \pm 15.30$ | |
| AdamW | 5e-5 | $0.8031 \pm 0.0021$ | $0.0077 \pm 0.0002$ | $652.92 \pm 5.26$ | **28.36** |
| PDT | | $0.8149 \pm 0.0037$ | $0.0013 \pm 0.0002$ | $467.76 \pm 6.05$ | |
| RMSprop | 0.0001 | $0.7996 \pm 0.0032$ | $4.1\text{e-}5 \pm 1.6\text{e-}5$ | $661.08 \pm 0.42$ | **15.35** |
| PDT | | $0.8108 \pm 0.0026$ | $2.4\text{e-}5 \pm 0.6\text{e-}5$ | $559.61 \pm 0.00$ | |
| Shampoo | 0.001 | $0.8012 \pm 0.0071$ | $0.0040 \pm 0.0005$ | $736.56 \pm 10.58$ | **16.03** |
| PDT | | $0.8101 \pm 0.0043$ | $0.0033 \pm 0.0003$ | $618.49 \pm 12.72$ | |
| LAMB | 0.001 | $0.8034 \pm 0.0025$ | $0.1215 \pm 0.0085$ | $663.25 \pm 5.44$ | **44.24** |
| PDT | | $0.8140 \pm 0.0027$ | $0.0036 \pm 0.0006$ | $369.82 \pm 10.59$ | |

Table 3: Performance comparison of SimSiam pre-training on CIFAR-10 with a ResNet-18 backbone, trained for 200 epochs, lr=0.03, batch size=256, momentum=0.9, with CosineAnnealingLR scheduler.

| Optimizer | Final Accuracy | TTB-Loss (s) | TTB-Acc (s) | Runtime Reduction (%) | |
|---|---|---|---|---|---|
| | | | | Train Loss | Val. Acc. |
| SGD Momentum | $0.7285 \pm 0.0166$ | $9611.35 \pm 837.98$ | $7353.12 \pm 1063.96$ | **48.78** | **16.10** |
| PDT | $0.7685 \pm 0.0144$ | $4922.92 \pm 712.55$ | $6169.04 \pm 1142.97$ | | |

In addition, further analysis of PDT's performance under non-i.i.d. training conditions is presented in Appendix A.11. The cross-domain evaluation on natural language processing tasks is in Appendix A.12.

## 4.2 MASKING STRATEGY: RANDOM SELECTION AND VALIDATION LOSS?

We study the effectiveness of the proposed masking strategy by comparing it with two ad hoc strategies. First, we perform a series of runs in which we randomly select subsets of weights and increase their learning rates. In Fig. 5, regions highlighted in green show results from different runs where subsets of weights had their learning rates increased to match the step number used in the predictions. The selection ratio used here matches the average masking ratio applied during PDT. The results indicate that randomly accelerating weights cannot match the performance improvements from PDT and often lead to significant instability during training.

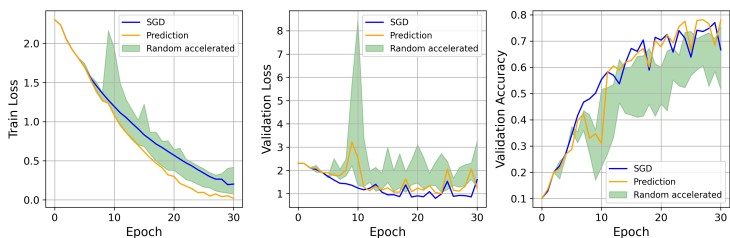

Figure 5: PDT vs. randomly selected subsets with higher learning rates (with the same mask ratio). Trained on CIFAR-10 using AlexNet, batch size=256, lr = 0.05.

Second, we perform a series of runs where subsets of Koopman predicted weights are randomly selected. The regions highlighted in green in Fig. 6 show the outcomes of these trials. Quite frequently, these runs result in gradient explosions, leading to non-recoverable errors (NaN values) in subsequent epochs. This experiment underscores the importance of a principled masking strategy in Koopman Training. Random masking, without considering the training dynamics, can lead to severe divergence and training failure. Our findings highlight that strategic selection based on "high-fidelity" predictions is crucial to the success of PDT. Furthermore, we conduct a comprehensive ablation study in Appendix A.9 to validate the necessity of combining both the acceleration effectiveness (Eq. 6) and dynamic consistency criteria (Eq. 7).

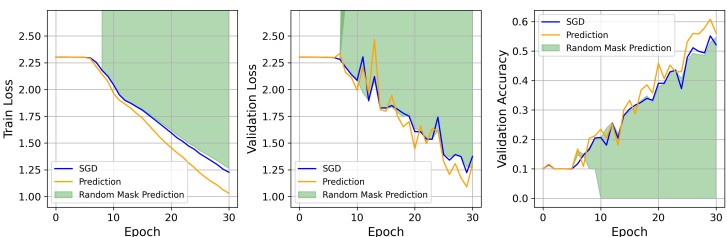

Figure 6: PDT vs. random mask prediction (with the same mask ratio). Trained on CIFAR-10 using AlexNet, batch size=256, lr = 0.01.

We implement another baseline scheduling scheme that switches between prediction and SGD based on the validation loss trend: apply prediction when validation loss decreases and roll back to SGD updates when validation loss starts to increase. Fig. 7 illustrates the training dynamics under this strategy. Initially, DMD is engaged due to its slight advantage in reducing validation loss. However, as training progresses, a significant surge in loss is observed, suggesting a misalignment between the DMD-predicted weights and the optimal trajectory for the network. Even after reverting to SGD, the model failed to recover, indicating that relying solely on validation loss as a trigger for switching between PDT and SGD is inadequate.

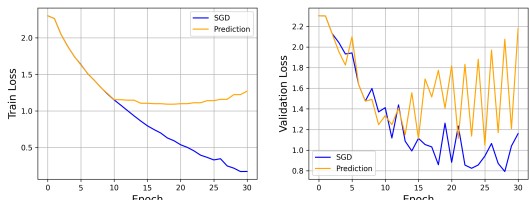

Figure 7: Performance comparison on CIFAR-10 using AlexNet: SGD vs. Koopman-based prediction (switching between prediction and SGD based on validation loss). L: Train loss. R: Validation loss.

## 5    CONCLUSION

This paper proposed a novel predictive differential training (PDT) framework based on the study of training dynamics. PDT incorporate the idea of "differential learning" into the predictive training framework for accelerated learning even for complex network structures. The key contribution is the design of an effective masking strategy based on a dynamic consistency analysis, which selects only those predicted weights of high-fidelity whose local training dynamics align with the global dynamics. Analogous to the saying *a rising tide lifts all boats*, in our setting, a subset of high-fidelity predicted weights facilitates more efficient training across the entire network!

The training process of a deep network with millions to billions of parameters indeed presents an intriguing dynamical system that the control community has not faced before. This would stimulate further investigation into the development of better data-driven dynamical system analysis algorithms in addition to DMD. Innovative approaches, such as streaming DMD (Hemati et al., 2014; Liew et al., 2022), can not only reduce the memory footprint of constructing trajectory matrices, but also improve computational efficiency.

ACKNOWLEDGMENTS

A portion of the computation for this work was performed on the University of Tennessee Infrastructure for Scientific Applications and Advanced Computing (ISAAC) computational resources. We also thank the anonymous reviewers for their constructive comments that significantly improved this paper.

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

## A APPENDIX

### A.1 CONVERGENCE PATH OF THE TOY EXAMPLE

To better illustrate the effectiveness of differential learning strategies, we designed a toy optimization problem with six variables. See Sec. 3.1 for the description of the problem. Starting from the initial point $[2.0, 2.0, 1.0, 0.5, -0.5, 1.5]$ with a learning rate of $0.01$. Fig. 8 shows the optimization trajectories in the x-y plane, where the background color represents the function value at each point. The blue line with dots represents the GD trajectory, while the red dashed line shows the path of accelerated GD where x, y, z variables use 3x learning rate. All points on the trajectories represent the state after each optimization step. The arrows indicate where each method reaches the threshold value (0.1). Building upon this observation, we apply our proposed PDT method to the same optimization problem. Fig. 9 presents the comparison between standard GD and our proposed PDT method on the same toy optimization problem. Fig. 9(a) uses the same visualization scheme as Fig. 8, showing how PDT follows a similar path but reaches the threshold faster (PDT reaches the threshold in 27 steps). Fig. 9(b) clearly demonstrates the acceleration effect, where PDT's loss decreases more rapidly than GD. The horizontal dashed line indicates the threshold value used as the stopping criterion.

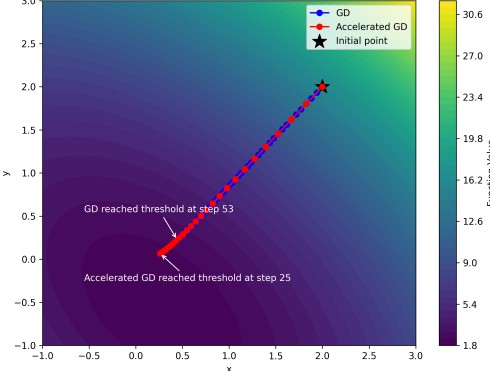

Figure 8: The differential learning trajectory of the toy example provided in Sec. 3.1. Only the $x$ and $y$ dimensions are shown.

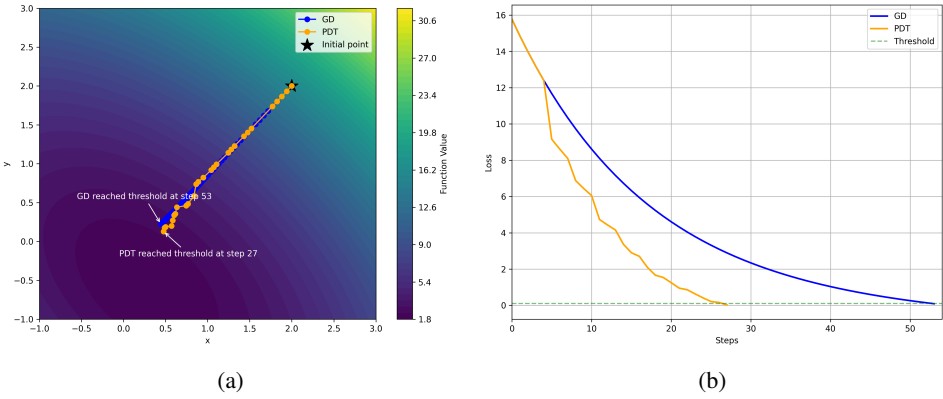

(a)          (b)

Figure 9: Performance comparison between GD (53 steps) and PDT (27 steps) on the toy optimization problem. (a) Optimization trajectories in the x-y plane. (b) Loss values during optimization.

## A.2 IMPLEMENTATION DETAILS OF PDT

### A.2.1 DERIVATION OF THE DMD ALGORITHM

In the main text, we introduced the Koopman operator approximation $A = W_{i+1}W_i^\dagger$. However, as noted, $A$ is an $N \times N$ matrix (where $N$ is the number of parameters, typically millions or billions), making direct computation intractable. In our implementation (see Appendix A.2.2), we employ the DMD algorithm (Tu et al., 2014) to compute the spectral decomposition of $A$ in a low-rank subspace. This section provides the step-by-step derivation linking the snapshot matrices $W_i, W_{i+1}$ in Eq. 4 to the spectral components $\Phi, \Lambda$ used in Eq. 5.

Let $\mathbf{w}_i \in \mathbb{R}^N$ be the flattened weight vector of the neural network at the current epoch $i$. To capture the training dynamics, we construct two snapshot matrices using the weight trajectories from the past $h$ epochs: *Input Matrix* (corresponding to $W_i$ in Eq. 4) contains the sequence of weights from the history buffer, excluding the last weight state; *Shifted Matrix* (corresponding to $W_{i+1}$ in Eq. 4) contains the same sequence shifted forward by one time step, ending with the current weight $\mathbf{w}_i$. For the remainder of this derivation, we refer to these matrices as $W_i$ and $W_{i+1}$ to maintain consistency with the main text. Both matrices are in $\mathbb{R}^{N \times (h-1)}$, where $h \ll N$ (e.g., $h = 5$).

We first compute the reduced SVD of $W_i$:

$$W_i \approx U\Sigma V^T \tag{8}$$

where $U \in \mathbb{R}^{N \times r}$, $\Sigma \in \mathbb{R}^{r \times r}$, and $V \in \mathbb{R}^{(h-1) \times r}$. Here $r \leq h$ is the truncation rank. This step reduces the dimensionality from the vast parameter space $N$ to the small snapshot space $r$.

Instead of computing $A = W_{i+1}W_i^\dagger = W_{i+1}V\Sigma^{-1}U^T$, we project the high-dimensional operator $A$ onto the low-dimensional subspace spanned by the proper orthogonal decomposition (POD) modes $U$ (Berkooz et al., 1993). We compute the proxy matrix $\tilde{A} \in \mathbb{R}^{r \times r}$:

$$\tilde{A} = U^T A U = U^T(W_{i+1}W_i^\dagger)U = U^T W_{i+1}V\Sigma^{-1} \tag{9}$$

Computationally, this involves multiplying the large matrix $W_{i+1}$ by small matrices $V$ and $\Sigma^{-1}$, then projecting onto $U$. This results in a tiny $r \times r$ matrix that captures the essential dynamics of the full system.

Since $\tilde{A}$ is small ($r \times r$), we can efficiently compute its eigendecomposition:

$$\tilde{A}\Psi = \Psi\Lambda \tag{10}$$

where $\Lambda = \text{diag}(\lambda_1, \ldots, \lambda_r)$ contains the eigenvalues (which approximate the eigenvalues of the full operator $A$), and $\Psi \in \mathbb{C}^{r \times r}$ contains the eigenvectors of $\tilde{A}$.

The eigenvectors $\Psi$ are in the low-dimensional subspace. To map the eigenvectors back to the full parameter space, we use the *Standard DMD* formulation, which is computationally efficient and numerically stable:

$$\Phi = U\Psi \tag{11}$$

This yields the DMD modes $\Phi \in \mathbb{C}^{N \times r}$. This step is crucial as it provides the mapping basis for our prediction equation. (Note: The *Exact DMD* formulation $\Phi = W_{i+1}V\Sigma^{-1}\Psi$ is also an option, but incurs additional computational cost).

With $\Phi$ and $\Lambda$ computed, we predict the future state $\tau$ steps ahead, starting from the current weight $\mathbf{w}_i$. The prediction equation (matching Eq. 5 in the main text) is derived as:

$$\mathbf{w}_{i+\tau}^{pred} = \text{Re}\{\Phi\Lambda^\tau\Phi^\dagger\mathbf{w}_i\} \tag{12}$$

Here, the term $\mathbf{c} = \Phi^\dagger\mathbf{w}_i$ represents the projection of the current weights onto the DMD modes (i.e., finding the mode amplitudes, which corresponds to the coefficients $c_k$ in Eq. 2). Since the modes $\Phi$ are generally non-orthogonal (as $A$ is not symmetric), $\Phi^\dagger$ denotes the Moore-Penrose pseudoinverse, which provides the least-squares solution $\mathbf{c} = \arg\min_{\tilde{\mathbf{c}}} |\mathbf{w}_i - \Phi\tilde{\mathbf{c}}|_2$. Finally, we take the real part of the result, as the neural network weights must be real-valued.

### A.2.2 ALGORITHM PSEUDOCODE

---

**Algorithm 1** PDT algorithm

---

**Require:** baseline optimizer $O_{\text{base}}$, past snapshot counts $h$, start epoch for prediction $T_0$, predicted steps $\tau$, prediction interval $T_i$, number of parameters $N$
**Ensure:** Trained model parameters $\mathbf{w}$
 1: Initialize weight history matrix $\mathbf{W}_{N \times h}$, counter $c_e = 0$
 2: **for** epoch $i = 0$ to $T$ **do**
 3:    **if** $i \geq T_0$ and $c_e \geq T_i$ **then**
 4:       Obtain $\mathbf{w}^{\text{opt}}(i-1)$ from history matrix $\mathbf{W}_{N \times h}$
 5:       Train model for one epoch using $O_{\text{base}}$, save weights after training as $\mathbf{w}^{\text{opt}}(i)$
 6:       Calculate DMD from $\mathbf{W}_{N \times h}$ to obtain modes $\Phi$ and eigenvalues $\Lambda$
 7:       **1) Decompose:** $\mathbf{c} \leftarrow \Phi^\dagger \mathbf{w}^{\text{opt}}(i)$ {Project current state onto modes}
 8:       **2) Evolve:** Compute future state $\mathbf{w}^{\text{pred}}(i+\tau) \leftarrow \text{Re}(\Phi \Lambda^\tau \mathbf{c})$
 9:       **3) Masking:** Create mask $M$ by comparing dynamics (Eqs. 6 and 7):
10:         SGD step: $\Delta_{sgd} = \mathbf{w}^{\text{opt}}(i) - \mathbf{w}^{\text{opt}}(i-1)$
11:         PDT step: $\Delta_{pdt} = \mathbf{w}^{\text{pred}}(i+\tau) - \mathbf{w}^{\text{opt}}(i)$
12:       **4) Assemble:** Update weights selectively
13:         $\mathbf{w}(i) \leftarrow M \odot \mathbf{w}^{\text{pred}}(i+\tau) + (1-M) \odot \mathbf{w}^{\text{opt}}(i)$
14:       Update model parameters with updated $\mathbf{w}(i)$
15:       $c_e \leftarrow 0$
16:    **else**
17:       Train model for one epoch using $O_{\text{base}}$
18:       $c_e \leftarrow c_e + 1$
19:    **end if**
20:    Update weight history matrix $\mathbf{W}_{N \times h}$
21: **end for**

---

## A.3 COMPUTATIONAL COMPLEXITY ANALYSIS

To provide a rigorous understanding of PDT's efficiency, we analyze its complexity in terms of both computation time and memory usage. We consider a DNN with $N$ parameters trained on a dataset with $S$ samples.

**Time Complexity.** The computational load for processing each batch using standard SGD is directly proportional to both the batch size ($B$) and the number of parameters ($N$), resulting in a complexity of $\mathcal{O}(B \times N)$ per batch, or $\mathcal{O}(S \times N)$ per epoch.

Integrating Koopman operator predictions into the DNN training process entails constructing a snapshot matrix from $h$ past epochs of the parameter trajectories, with the matrix dimension being $N \times h$. As derived explicitly in Appendix A.2.1, the prediction process involves several steps. The dominant operations and their complexities are:

- SVD of $W_i$: $\mathcal{O}(N \times h^2)$
- Computing $\tilde{A}$: $\mathcal{O}(N \times h^2)$
- Eigendecomposition of $\tilde{A}$: $\mathcal{O}(h^3)$
- Computing Modes $\Phi$: $\mathcal{O}(N \times h^2)$
- Prediction (Solve $\Phi^\dagger$): $\mathcal{O}(N \times h^2)$

The total complexity is $\mathcal{O}(N \times h^2)$. Since $h$ is a small constant (e.g., $h = 5$) in our experiments while $N$ can reach millions or even billions, the quadratic impact of $h$ remains manageable relative to $N$. Since the prediction step occurs only once per epoch (or every few epochs, depending on $T_i$), the amortized cost is minimal compared to the $\mathcal{O}(S \times N)$ cost of the baseline optimization over the full dataset.

**Space Complexity.** The additional memory requirement for PDT is dominated by the storage of the weight history matrix $\mathbf{W}$, which stacks $h$ snapshots of the model parameters. Thus, the space complexity is $\mathcal{O}(N \times h)$. For our default setting of $h = 5$, this corresponds to storing 5 additional copies of the model weights. On modern training hardware (e.g., NVIDIA A100 with 40GB+ VRAM),

this overhead is manageable. For instance, a ResNet-50 model ($N \approx 25.5M$) requires approximately 100 MB per snapshot (in float32 precision), totaling $\sim 500$ MB for $h = 5$, which is minor compared to the memory consumed by activation maps and optimizer states. Note that these snapshots are stored only temporarily and are overwritten after each DMD computation, so the space cost does not accumulate over training.

A detailed runtime and memory profiling analysis is provided in Appendix A.8.

### A.4 CONVERGENCE ANALYSIS

We analyze the convergence of the hybrid update that combines DMD-based predictions with gradient-descent updates under the masking strategy in Sec. 3.2. Throughout this section, we use $w_i$ to denote the current parameters and $\mathcal{L}(w)$ to denote the loss.

#### A.4.1 ASSUMPTIONS AND UPPER-BOUND CONSTRAINT

Assume that $\mathcal{L} : \mathbb{R}^d \to \mathbb{R}$ has $L$–Lipschitz continuous gradients, i.e.,

$$\|\nabla\mathcal{L}(x) - \nabla\mathcal{L}(y)\| \leq L\|x - y\|, \qquad \forall x, y \in \mathbb{R}^d.$$

Let $\mathbf{w}_i^{\text{opt}}$ denote the parameters after one step of the baseline optimizer

$$\mathbf{w}_{i+1}^{\text{opt}} = \mathbf{w}_i^{\text{opt}} - \eta\nabla\mathcal{L}(\mathbf{w}_i^{\text{opt}}),$$

with step size $\eta > 0$. Let $\mathbf{w}_{i+\tau}^{\text{pred}}$ denote the $\tau$–step DMD prediction starting from $\mathbf{w}_i^{\text{opt}}$. As Eq. 6 in Sec. 3.2, PDT enforces the magnitude bound

$$\|\mathbf{w}_{i+\tau}^{\text{pred}} - \mathbf{w}_i^{\text{opt}}\| \leq \tau\eta\|\nabla\mathcal{L}(\mathbf{w}_i^{\text{opt}})\|. \tag{13}$$

In addition, for each element $j$ accepted by the mask, the dynamic consistency criterion (Eq. 7) ensures the sign alignment, $sign(w_{i+k,j}^{\text{pred}} - w_{i+k-1,j}^{\text{pred}}) = sign(w_{i+1,j}^{\text{opt}} - w_{i,j}^{\text{opt}})$, where $k = 1, \cdots, \tau$. Hence, we have

$$sign(w_{i+\tau,j}^{\text{pred}} - w_{i,j}^{\text{opt}}) = sign(w_{i+1,j}^{\text{opt}} - w_{i,j}^{\text{opt}}) = sign(-\eta\nabla\mathcal{L}(w_{i,j}^{\text{opt}}))$$

This leads to

$$\left(w_{i+\tau,j}^{\text{pred}} - w_{i,j}^{\text{opt}}\right)\left(\nabla\mathcal{L}(w_{i,j}^{\text{opt}})\right) \leq 0.$$

#### A.4.2 UPDATE RULE WITH MASKING

Let $\mathbf{m}_i \in \{0, 1\}^N$ denote the binary mask. PDT forms the next iterate by

$$\mathbf{w}_{i+1} = \mathbf{m}_i \odot \mathbf{w}_{i+\tau}^{\text{pred}} + (\mathbf{1}^N - \mathbf{m}_i) \odot \mathbf{w}_{i+1}^{\text{opt}}, \tag{14}$$

where $\odot$ denotes elementwise multiplication. Define

$$\mathbf{d}_i := \mathbf{w}_{i+1} - \mathbf{w}_i^{\text{opt}}.$$

Elementwise,

$$d_{i,j} = \begin{cases} w_{i+\tau,j}^{\text{pred}} - w_{i,j}^{\text{opt}}, & m_{i,j} = 1, \\ -\eta\,\nabla\mathcal{L}(w_{i,j}^{\text{opt}}), & m_{i,j} = 0. \end{cases} \tag{15}$$

For masked elements, $|d_{i,j}| = |w_{i+\tau,j}^{\text{pred}} - w_{i,j}^{\text{opt}}| \leq \tau\eta|\nabla\mathcal{L}(w_{i,j}^{\text{opt}})|$ by Eq. 13. For unmasked ones, $|d_{i,j}| = \eta|\nabla\mathcal{L}(w_{i,j}^{\text{opt}})| \leq \tau\eta|\nabla\mathcal{L}(w_{i,j}^{\text{opt}})|$ since $\tau \geq 1$. Hence $|d_{i,j}| \leq \tau\eta|\nabla\mathcal{L}(w_{i,j}^{\text{opt}})|$ for all $j$, and

$$\|\mathbf{d}_i\| \leq \tau\eta\|\nabla\mathcal{L}(\mathbf{w}_i^{\text{opt}})\|. \tag{16}$$

Furthermore, for elements where $m_{i,j} = 1$, the acceleration effectiveness criterion (Eq. 6 lower bound) implies $|d_{i,j}| > \eta|\nabla\mathcal{L}_j|$. Combined with the sign consistency,

$$\nabla\mathcal{L}(w_{i,j}^{\text{opt}})\,d_{i,j} \leq -\eta(\nabla\mathcal{L}(w_{i,j}^{\text{opt}}))^2,$$

for all $j$. Summing,

$$\langle\nabla\mathcal{L}(\mathbf{w}_i^{\text{opt}}), \mathbf{d}_i\rangle \leq -\eta\|\nabla\mathcal{L}(\mathbf{w}_i^{\text{opt}})\|^2. \tag{17}$$

### A.4.3 DESCENT INEQUALITY AND CONVERGENCE

Because $\mathcal{L}$ has $L$–Lipschitz gradients, the standard descent lemma yields

$$\mathcal{L}(\mathbf{x} + \mathbf{d}) \ \leq \ \mathcal{L}(\mathbf{x}) + \langle \nabla \mathcal{L}(\mathbf{x}), \mathbf{d} \rangle + \frac{L}{2} \|\mathbf{d}\|^2. \tag{18}$$

Applying Eq. 18 with $\mathbf{x} = \mathbf{w}_i^{\mathrm{opt}}$ and $\mathbf{d} = \mathbf{d}_i$, and using Eq. 16– 17, we obtain

$$\mathcal{L}(\mathbf{w}_{i+1}) \leq \mathcal{L}(\mathbf{w}_i^{\mathrm{opt}}) - \eta \|\nabla \mathcal{L}(\mathbf{w}_i^{\mathrm{opt}})\|^2 + \frac{L}{2} \tau^2 \eta^2 \|\nabla \mathcal{L}(\mathbf{w}_i^{\mathrm{opt}})\|^2. \tag{19}$$

Grouping the terms,

$$\mathcal{L}(\mathbf{w}_{i+1}) \leq \mathcal{L}(\mathbf{w}_i^{\mathrm{opt}}) - \eta \left( 1 - \frac{L \eta \tau^2}{2} \right) \|\nabla \mathcal{L}(\mathbf{w}_i^{\mathrm{opt}})\|^2. \tag{20}$$

A sufficient condition for the coefficient, $1 - \frac{L \eta \tau^2}{2}$, to be positive is

$$\eta \leq \frac{2}{L \tau^2}. \tag{21}$$

Under Eq. 21, we obtain the GD-style descent inequality

$$\mathcal{L}(\mathbf{w}_{i+1}) \ \leq \ \mathcal{L}(\mathbf{w}_i^{\mathrm{opt}}). \tag{22}$$

Thus $\{\mathcal{L}(\mathbf{w}_i)\}$ is monotonically non-increasing and bounded below, hence convergent. Standard results for gradient-based methods with bounded steps and $L$-smooth losses (e.g., inexact gradient descent) then imply that

$$\|\nabla \mathcal{L}(\mathbf{w}_i)\| \to 0,$$

and every limit point of $\{\mathbf{w}_i\}$ is stationary.

### A.5 DETAILED TRAINING CURVES ON VARIOUS NETWORK STRUCTURES

Fig. 10 presents the detailed training curves and performance comparison of PDT and baseline optimizer across various network structures.

### A.6 EFFECT OF TRAINING HYPERPARAMETERS

Several primary hyperparameters require careful consideration in PDT:

**Prediction Steps ($\tau$):** Derived from DMD, the number of prediction steps significantly influences the training speed. As shown in Fig. 11(a), training accelerates within a certain range of prediction steps. However, extending beyond a critical threshold, such as nine steps in our study, can introduce large errors and potentially cause gradient explosion.

**Prediction Interval ($T_i$):** The interval between Prediction blocks impacts the effectiveness of acceleration, as depicted in Fig. 11(b). A shorter interval can enhance training speed if the predictions are accurate. Nevertheless, the quality of predictions may decline as the training progresses, rendering the network more sensitive to errors, particularly as it nears convergence.

**Starting Epoch ($T_0$):** The starting epoch for acceleration must be greater than or equal to the number of epochs used to build the snapshot, as illustrated in Fig. 11(c). The initiation of acceleration is influenced by factors such as initialization, learning rate, and model architecture.

**Past Snapshot Counts ($h$):** Fig. 11(d) indicates that the number of epochs needed to construct the snapshot matrix for prediction also influences the train loss. This value cannot be too small or too large. If it is too small, the snapshot will not have sufficient measurements to precisely estimate the dynamics of the training process. If it is too large, DMD would have missed the local dynamics with only a coarser grasp of the general training dynamics.

Overall, these PDT-related hyperparameters are robust across optimizers, architectures, and datasets. Based on our experience, below are "Rule of Thumb" guidelines to help find the appropriate hyperparameters for different scenarios.

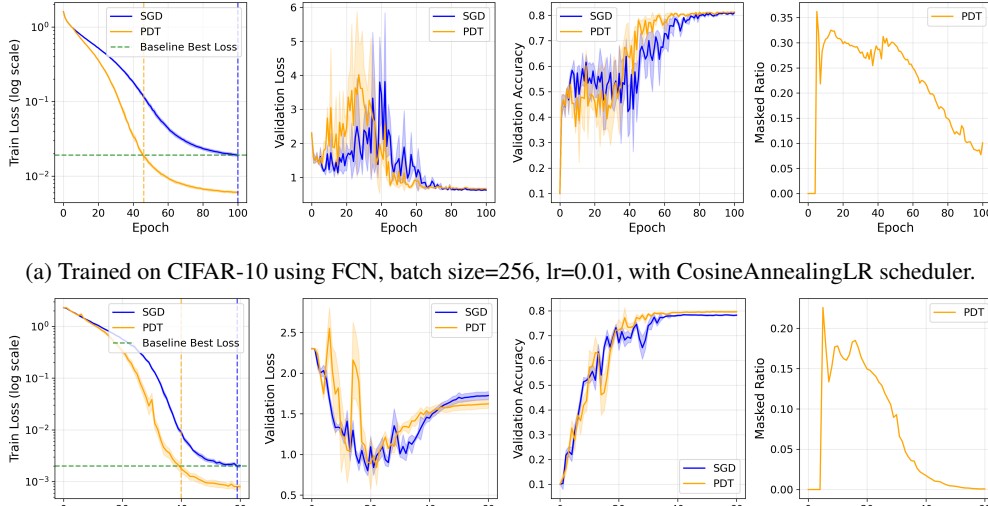

(a) Trained on CIFAR-10 using FCN, batch size=256, lr=0.01, with CosineAnnealingLR scheduler.

(b) Trained on CIFAR-10 using AlexNet, batch size=256, lr=0.05, with CosineAnnealingLR scheduler.

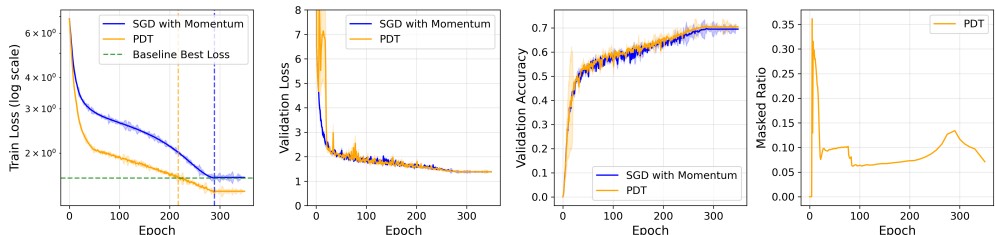

(c) Trained on ImageNet-1K using ResNet-50, batch size=1800, lr=0.1, momentum=0.9, with CosineAnnealingLR scheduler.

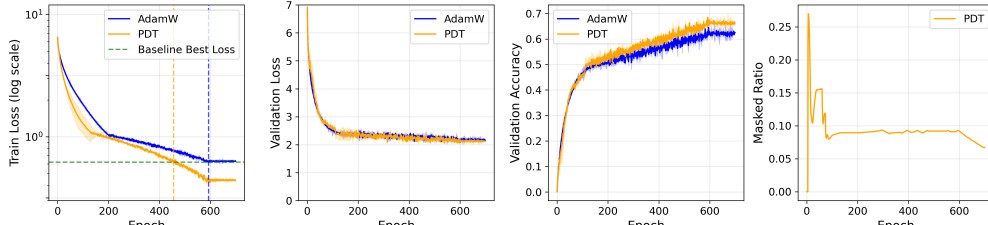

(d) Trained on ImageNet-1K using ViT-Base, batch size=1800, lr=0.003, momentum=0.9, with CosineAnnealingLR scheduler.

Figure 10: Performance comparison between baseline optimization and PDT.

- Past Snapshot counts ($h$): $h = 5$ is a "sweet spot". Smaller $h$ is insufficient for capturing dynamics, larger $h$ includes "stale" weights from much earlier training and introduces additional overhead. For networks with extremely dynamic changes, a larger value of $h$ (e.g., $h$=10) is also worth trying.

- Prediction Steps ($\tau$): We recommend starting with $\tau = 5$ as a robust default value. This value provides a good balance between acceleration benefit and prediction accuracy across diverse architectures and datasets. Users can increase $\tau$ to 7 if their training loss curves are very stable and exhibit minimal variance. Conversely, if gradient explosion or instability occurs, reducing $\tau$ to 3 provides a more conservative acceleration while maintaining stability. A practical configuration is to set $\tau \in [3, 7]$. Too large leads to divergence, while too small makes it meaningless.

- Prediction Interval ($T_i$): We recommend setting $T_i = 1$ as the default. If the training process is unstable, then gradually increase the interval.

- Start Epoch ($T_0$): The start epoch $T_0$ should typically be equal to $h$ to ensure sufficient history is available for the first prediction.

To thoroughly evaluate the effectiveness and robustness of PDT under different training configurations, we conduct comprehensive experiments across different learning rates from 0.001 to 0.1 (0.001, 0.01, 0.05, 0.1) and batch sizes from 32 to 512 (32, 64, 128, 256, 512). All experiments were repeated with five random seeds (0, 100, 200, 300, 400) to ensure statistical significance. All experiments are performed on AlexNet with the CIFAR-10 dataset, using SGD as the baseline optimizer and trained for 60 epochs. The PDT-related hyperparameters mentioned in Sec. A.6 were set to prediction step=5, prediction interval=1, start epoch=5, and past snapshot counts=5.

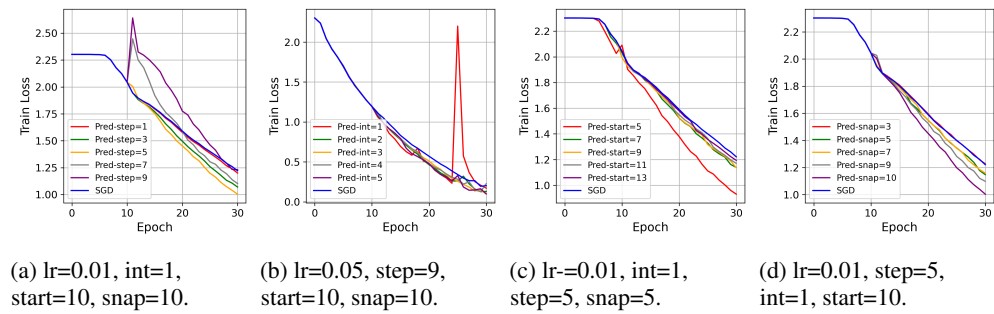

(a) lr=0.01, int=1, start=10, snap=10.

(b) lr=0.05, step=9, start=10, snap=10.

(c) lr-=0.01, int=1, step=5, snap=5.

(d) lr=0.01, step=5, int=1, start=10.

Figure 11: The influence of different parameters. (a) prediction steps, (b) prediction interval, (c) starting epoch, (d) past snapshot counts. Trained on CIFAR-10 using AlexNet, batch size=256.

The results in Table 4 show the impact of different batch sizes and learning rates on the performance of PDT. At lower learning rates (0.001, 0.01, and 0.05), PDT consistently outperforms SGD in terms of convergence speed across different batch sizes. PDT shows a significant reduction in the runtime to reach baseline best loss, with an average runtime reduction of 22.76% compared to SGD. For higher learning rates (0.1), both SGD and PDT struggled to achieve stable training, and PDT's advantage over SGD became less pronounced. Sometimes PDT can significantly reduce the convergence time (for example, when batch size = 64), but other times the accuracy will drop significantly after reaching a high point, or even result in gradient explosion. This suggests that the high learning rate introduced significant stochasticity, reducing the effectiveness of PDT's prediction mechanism. Smaller batch sizes (32, 64) generally achieve more significant runtime reductions.

To address the stability issues observed at higher learning rates and larger batch sizes, different from the previous fixed learning rate, we investigated the effectiveness of the learning rate scheduler. We tested the Cosine Annealing learning rate scheduler with a minimum learning rate of 1e-3. Taking batch size 256 as an example, we observe significantly improved stability and performance. The results are shown in Table 5. The results are particularly noteworthy at higher learning rates (lr=0.1), where the previous experiments in Table 4 show considerable variance. With the cosine annealing scheduler, PDT achieves consistent accuracy improvements across all learning rates while maintaining substantial runtime reductions.

To further investigate PDT's compatibility with different optimization methods, we compare its performance when integrated with different optimizers (SGD, SGD with momentum, and Adam) while keeping the network architecture and other configurations fixed. For SGD with momentum, we set the momentum factor to 0.9. All experiments are conducted on AlexNet with CIFAR-10 using batch size 256, maintaining the same PDT hyperparameters as in previous experiments. The learning rate is 0.1 for SGD, 0.001 for SGD with Momentum, 0.0005 for Adam. The results are shown in Table 6.

Table 4: Impact of learning rates and batch sizes on PDT performance. Trained on CIFAR-10 using AlexNet. Note: bold numbers indicate the best performance and underlined numbers indicate the second best performance for each column.

| Batch Size | lr | Method | Final Accuracy (mean ± std) | Best Train Loss (mean ± std) | Time to Baseline Best Loss (s) (mean ± std) | Runtime Reduction (%) |
|---|---|---|---|---|---|---|
| 32 | 0.001 | SGD | 0.6981 ± 0.0458 | 0.6376 ± 0.0127 | 1232.29 ± 4.45 | 40.64 |
| | | PDT | 0.6903 ± 0.0885 | 0.2724 ± 0.0166 | 731.52 ± 12.84 | |
| | 0.01 | SGD | 0.8118 ± 0.0041 | 0.0046 ± 0.0008 | 1194.89 ± 21.09 | 24.25 |
| | | PDT | **0.8146 ± 0.0048** | 0.0021 ± 0.0012 | 905.07 ± 120.51 | |
| | 0.05 | SGD | 0.8049 ± 0.0053 | 0.0156 ± 0.0029 | 1180.72 ± 12.31 | **64.57** |
| | | PDT | 0.8020 ± 0.0052 | 0.0149 ± 0.0073 | 418.38 ± 0.00 | |
| | 0.1 | SGD | 0.1000 ± 0.0000 | 0.3346 ± 0.0098 | 1172.49 ± 39.08 | - |
| | | PDT | 0.1000 ± 0.0000 | 0.3364 ± 0.0132 | - | |
| 64 | 0.001 | SGD | 0.5384 ± 0.0173 | 1.2295 ± 0.0261 | 902.16 ± 19.68 | 35.82 |
| | | PDT | 0.5329 ± 0.1152 | 0.8798 ± 0.0257 | 578.99 ± 55.74 | |
| | 0.01 | SGD | 0.7850 ± 0.0226 | 0.0087 ± 0.0030 | 800.35 ± 5.39 | 23.32 |
| | | PDT | 0.8140 ± 0.0021 | 0.0015 ± 0.0010 | 613.70 ± 8.80 | |
| | 0.05 | SGD | 0.8067 ± 0.0035 | 0.0051 ± 0.0016 | 798.20 ± 3.50 | 27.54 |
| | | PDT | 0.8029 ± 0.0029 | 0.0045 ± 0.0006 | 578.36 ± 16.48 | |
| | 0.1 | SGD | 0.6442 ± 0.2733 | 0.0484 ± 0.0522 | 910.37 ± 18.03 | 56.23 |
| | | PDT | 0.7976 ± 0.0033 | 0.0218 ± 0.0011 | **398.48 ± 21.34** | |
| 128 | 0.001 | SGD | 0.2882 ± 0.0212 | 1.8456 ± 0.0300 | 812.42 ± 21.20 | 17.48 |
| | | PDT | 0.2951 ± 0.0440 | 1.6972 ± 0.0272 | 670.37 ± 23.93 | |
| | 0.01 | SGD | 0.7825 ± 0.0065 | 0.0675 ± 0.0052 | 661.09 ± 6.35 | 14.68 |
| | | PDT | 0.8009 ± 0.0062 | 0.0058 ± 0.0008 | 564.02 ± 16.35 | |
| | 0.05 | SGD | 0.7969 ± 0.0093 | 0.0039 ± 0.0017 | 662.48 ± 7.73 | 9.15 |
| | | PDT | 0.8011 ± 0.0067 | 0.0016 ± 0.0017 | 601.86 ± 17.78 | |
| | 0.1 | SGD | 0.7916 ± 0.0027 | 0.0083 ± 0.0014 | 803.93 ± 3.07 | 8.20 |
| | | PDT | 0.7863 ± 0.0087 | 0.0096 ± 0.0016 | 737.97 ± 0.00 | |
| 256 | 0.001 | SGD | 0.1171 ± 0.0092 | 2.2991 ± 0.0011 | 747.83 ± 20.30 | 7.08 |
| | | PDT | 0.1453 ± 0.0213 | 2.2979 ± 0.0026 | 694.91 ± 14.63 | |
| | 0.01 | SGD | 0.6989 ± 0.0301 | 0.5814 ± 0.0147 | 660.37 ± 0.71 | 19.98 |
| | | PDT | 0.7450 ± 0.0236 | 0.1855 ± 0.0172 | 528.41 ± 7.26 | |
| | 0.05 | SGD | 0.7931 ± 0.0034 | **0.0004 ± 0.0003** | 648.39 ± 8.57 | 21.71 |
| | | PDT | 0.7916 ± 0.0016 | 0.0015 ± 0.0014 | 507.62 ± 11.36 | |
| | 0.1 | SGD | 0.3742 ± 0.3359 | 0.0508 ± 0.0576 | 771.77 ± 3.06 | - |
| | | PDT | 0.3796 ± 0.3425 | 0.0012 ± 0.0011 | - | |
| 512 | 0.001 | SGD | 0.1170 ± 0.0251 | 2.3017 ± 0.0005 | 748.44 ± 42.46 | 6.23 |
| | | PDT | 0.1377 ± 0.0288 | 2.3020 ± 0.0001 | 701.82 ± 23.31 | |
| | 0.01 | SGD | 0.5710 ± 0.0203 | 1.1920 ± 0.0238 | 671.28 ± 9.03 | 18.89 |
| | | PDT | 0.5985 ± 0.0078 | 0.8311 ± 0.0252 | 544.46 ± 12.10 | |
| | 0.05 | SGD | 0.7717 ± 0.0038 | 0.0311 ± 0.0174 | 668.59 ± 7.30 | 10.11 |
| | | PDT | 0.7669 ± 0.0237 | 0.0034 ± 0.0014 | 601.01 ± 44.11 | |
| | 0.1 | SGD | 0.3721 ± 0.3332 | 0.0648 ± 0.0735 | 768.97 ± 3.12 | - |
| | | PDT | 0.4420 ± 0.3420 | 0.0373 ± 0.0155 | - | |

Table 5: Impact of learning rates on PDT performance. Trained on CIFAR-10 using AlexNet, batch size=256, with CosineAnnealingLR scheduler, minimum learning rate 1e-3. Note: bold numbers indicate the best performance and underlined numbers indicate the second best performance for each column.

| Batch Size | lr | Method | Final Accuracy (mean ± std) | Best Train Loss (mean ± std) | Time to Baseline Best Loss (s) (mean ± std) | Runtime Reduction (%) |
|---|---|---|---|---|---|---|
| | 0.001 | SGD | 0.1217 ± 0.0126 | 2.2991 ± 0.0011 | 757.66 ± 26.54 | 9.88 |
| | | PDT | 0.1461 ± 0.0213 | 2.2980 ± 0.0025 | 682.79 ± 2.13 | |
| 256 | 0.01 | SGD | 0.6451 ± 0.0102 | 0.9276 ± 0.0212 | 745.97 ± 47.19 | **41.54** |
| | | PDT | 0.6974 ± 0.0073 | 0.5853 ± 0.0159 | 436.07 ± 16.09 | |
| | 0.05 | SGD | 0.7852 ± 0.0016 | 0.0020 ± 0.0001 | 675.04 ± 27.56 | 37.13 |
| | | PDT | 0.7936 ± 0.0030 | 0.0006 ± 0.0001 | **424.39 ± 20.40** | |
| | 0.1 | SGD | 0.7930 ± 0.0023 | **0.0002 ± 0.0000** | 665.27 ± 9.08 | 19.67 |
| | | PDT | **0.7978 ± 0.0032** | **0.0002 ± 0.0000** | 534.41 ± 12.64 | |

Table 6: Impact of baseline optimizers (SGD, SGD with Momentum, and Adam) on PDT performance. Trained on CIFAR-10 using AlexNet, batch size=256, momentum=0.9, with CosineAnnealingLR scheduler. Note: bold numbers indicate the best performance and underlined numbers indicate the second best performance for each column.

| lr | Method | Final Accuracy (mean ± std) | Best Train Loss (mean ± std) | Time to Baseline Best Loss (s) (mean ± std) | Runtime Reduction (%) |
|---|---|---|---|---|---|
| 0.1 | SGD | 0.7930 ± 0.0023 | 0.0002 ± 0.0000 | 665.27 ± 9.08 | 19.67 |
| | PDT | 0.7978 ± 0.0032 | 0.0002 ± 0.0000 | 534.41 ± 12.64 | |
| 0.001 | Momentum | 0.6672 ± 0.0068 | 0.8609 ± 0.0166 | 752.74 ± 9.62 | **41.06** |
| | PDT | 0.7298 ± 0.0051 | 0.5358 ± 0.0165 | **443.68 ± 8.75** | |
| 0.0005 | Adam | 0.7952 ± 0.0063 | **0.0001 ± 0.0000** | 779.13 ± 11.81 | 14.87 |
| | PDT | **0.8050 ± 0.0050** | 0.0002 ± 0.0000 | 663.28 ± 15.30 | |

## A.7 ANALYSIS OF MASK DISTRIBUTION

We further analyze the mask distribution and dynamics in AlexNet. Fig. 12 shows how the ratio of the predicted weights evolves over training epochs. The analysis is conducted using the same experimental setup as in Fig. 10(b), where AlexNet is trained on CIFAR-10.

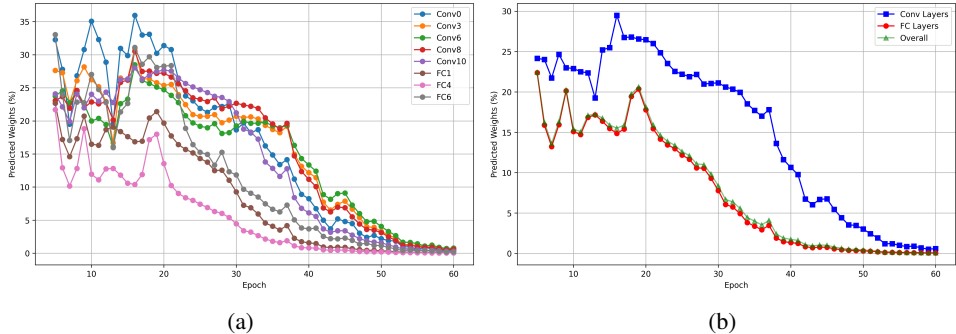

(a)                                                                (b)

Figure 12: Analysis of mask distribution in AlexNet. (a) Layer-wise mask evolution over training epochs. (b) Comparison of prediction ratios between convolutional and fully connected layers.

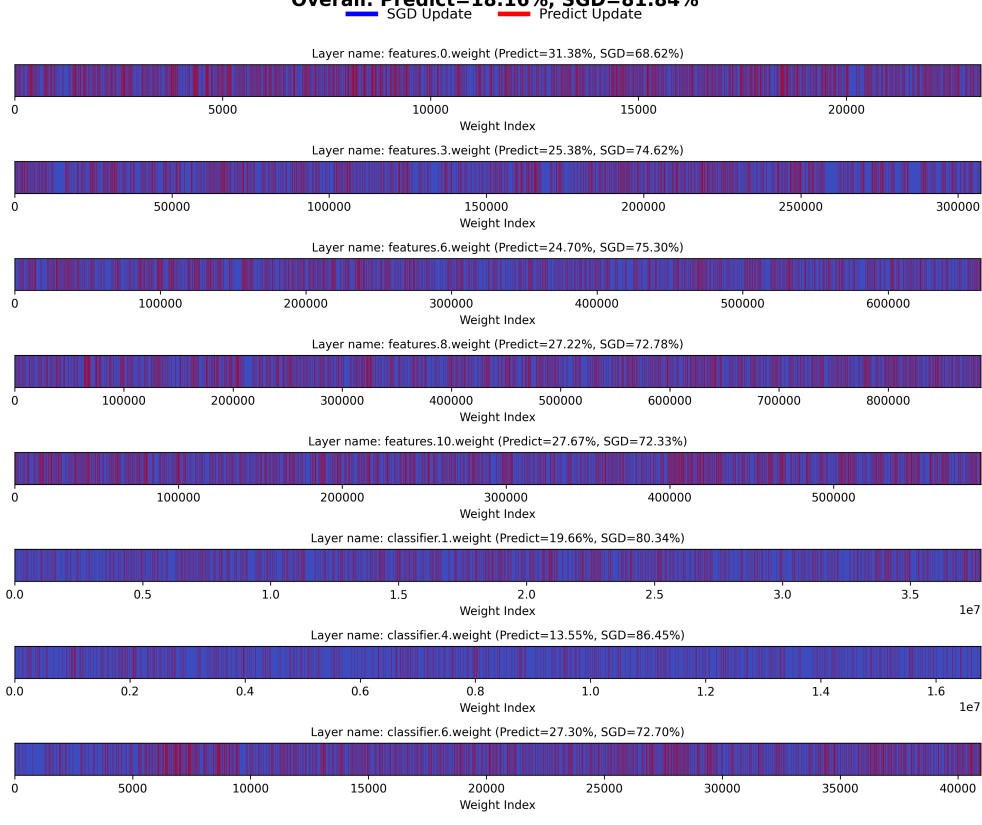

Figure 13: A snapshot at epoch 20 with the mask heatmap for different layers.

Fig. 12(a) presents the layer-wise evolution of the ratio of the predicted weights throughout the training process. We observe a pattern here: the masked ratio of each layer starts relatively high, maintaining a stable period, and then gradually declining. The decline phase at the later epochs suggests that as the network approaches convergence, it relies more on gradient-based updates rather than predictions. This aligns with the intuition that predictive updates can be beneficial in the early

phases for accelerating convergence but become less necessary as the model stabilizes. The early convolutional layers (e.g., Conv0) exhibit more fluctuations in the percentage of predictive updates, suggesting a higher sensitivity to training dynamics.

Fig. 12(b) tracks the evolution of predicted weights ratios by layer type. The overall percentage of predictively updated weights is also included. Interestingly, convolutional layers consistently maintain a higher prediction ratio compared to fully connected layers throughout the training process. Due to the majority of the weights in the AlexNet network belonging to the fully connected layers (54.6 million vs. 2.5 million), the overall masked ratio closely follows the trend of fully connected layers.

To provide a finer-grained visualization of the mask distribution, Fig. 13 depicts the mask heatmap for different layers at epoch 20. Each horizontal band represents a layer, where blue regions indicate weights updated by SGD, and red regions correspond to weights updated by prediction results. We can observe that the distribution of predictive updates is not uniform across layers, with some layers showing clustered regions of predictive updates, potentially indicating structured weight adaptations.

### A.8 PROFILING AND OVERHEAD ANALYSIS

To validate our theoretical complexity analysis and quantitatively measure the overhead, we conduct comprehensive profiling experiments across three diverse architectures: AlexNet (CNN, 57M parameters) on CIFAR-10, ResNet-50 (deep CNN, 25.6M parameters) on ImageNet-1k, and ViT-Base (Transformer, 86.4M parameters) on ImageNet-1k. We measure wall-clock time, GPU memory usage, and FLOPs for both the baseline optimizer and PDT.

#### A.8.1 MEASUREMENT METHODOLOGY

**Memory Overhead.** The *Peak GPU memory* captures true peak GPU usage during training, measured via `torch.cuda.max_memory_allocated()`. *Currently Allocated* is the memory actively allocated at the end of each epoch. *Reserved by Allocator* is the total memory reserved by PyTorch's caching allocator. The *PDT Memory Overhead* consists of *Snapshot Storage* ($h$ copies of weights), *SVD Workspace* (the temporary storage for intermediate matrices), and *Other* (includes optimizer momentum buffers, weight cloning buffers, temporary tensors, and memory fragmentation). The *PDT memory overhead ratio* is computed as the total PDT overhead divided by the baseline's peak GPU memory.

**Runtime Overhead.** The *Total PDT Runtime Overhead* is defined as the difference between the PDT epoch time and the baseline SGD epoch time. We further decompose this into *Core PDT Operations* (including SVD decomposition, DMD-based weight prediction, and gradient masking) and *Auxiliary Operations* (including weight cloning for snapshot storage, CUDA synchronization, and data movement between CPU/GPU). The *PDT runtime overhead ratio* is computed as total PDT overhead divided by total epoch time.

**FLOPs Overhead.** Total floating-point operations for the entire training process.

All profiling experiments are conducted on a single GPU (Nvidia RTX A6000 or H100) to ensure accurate measurement of peak memory and timing without the interference of distributed communication overheads. For PDT, we use a past snapshot counts of $h = 5$ and a prediction interval of $T_i = 1$.

#### A.8.2 PROFILING RESULTS

**1. AlexNet on CIFAR-10.** Table 7 summarizes the results for AlexNet on CIFAR-10. The peak GPU memory usage for PDT increases by approximately $8.4$ GB compared to the baseline. This absolute increase is primarily due to the snapshot storage (1.1 GB for $h = 5$) and the SVD workspace (1.7 GB for intermediate matrices). The remaining overhead ($\sim 5.6$ GB) is attributed to temporary buffers and memory fragmentation during the weight update process. While the relative increase (1145%) appears large due to the small baseline footprint of AlexNet on CIFAR-10, the absolute peak usage (9.1 GB) fits comfortably within the capacity of modern GPUs. This confirms that the $\mathcal{O}(N \times h)$ space complexity is practical, as described in Appendix A.3. The total PDT overhead is approximately 333 ms per epoch, with the PDT overhead ratio for only **4.30%**. Decomposing this overhead reveals

that core PDT operations account for 194 ms (58.3%), while auxiliary operations (weight cloning and synchronization) account for 138.8 ms (41.7%). Notably, the SVD operation itself takes only $\sim 98$ ms, empirically confirming our theoretical claim that the SVD cost is computationally efficient even for frequent predictions. The additional FLOPs introduced by PDT are negligible. The total training FLOPs increase from $3.4043 \times 10^{15}$ (baseline) to $3.4047 \times 10^{15}$ (PDT), a 0.012% increase, validating that the computational cost is dominated by gradient calculations in the baseline optimizer rather than in the PDT operations.

Table 7: Memory and Runtime overhead of PDT compared to baseline. Trained on CIFAR-10 using AlexNet, batch size=256, SGD baseline optimizer with lr=0.05, with CosineAnnealingLR scheduler. All experiments ran on a single Nvidia RTX A6000 (48 GB) GPU.

| Metric | Baseline | PDT |
|---|---|---|
| *GPU Memory Usage* | | |
|     Peak GPU Memory (MB) | 735.9 | 9158.2 |
|     Currently Allocated (MB) | 468.7 | 5078.0 |
|     Reserved by Allocator (MB) | 994.0 | 11454.0 |
| *PDT Memory Overhead* | | |
|     Total PDT Overhead (MB) | — | 8422.4 |
|     Overhead Ratio (%) | — | **1145%** |
| *PDT Memory Overhead Breakdown* | | |
|     Snapshot Storage (MB) | — | 1088.0 |
|     SVD Workspace (MB) | — | 1740.9 |
|     Other (MB) | — | 5593.4 |
| *PDT Runtime Overhead (per epoch)* | | |
|     Avg Time per Epoch (s) | 7.465 | 7.746 |
|     Total PDT Overhead (s) | — | 0.333 |
|     PDT Overhead Ratio (%) | — | **4.30%** |
| *PDT Runtime Overhead Breakdown* | | |
|     **Core PDT Operations (ms)** | — | 194.0 |
|       ↪ SVD decomposition (ms) | — | 98.3 |
|       ↪ Prediction (ms) | — | 88.1 |
|       ↪ Masking (ms) | — | 7.6 |
|     Auxiliary Operations (ms) | — | 138.8 |

**2. ResNet-50 on ImageNet-1k.** To further validate overhead analysis, we conducted comprehensive profiling on ResNet-50 + ImageNet-1k with varying batch sizes (64, 128, 256). Table 8 presents detailed memory and runtime measurements for both baseline and PDT. The absolute total PDT memory overhead remains nearly constant ($\sim 1.46$ GB) across all batch sizes, validating that PDT's space complexity $\mathcal{O}(N \times h)$ is independent of data scale. Consequently, the PDT memory overhead ratio decreases monotonically as baseline memory increases with batch size from 47.9% (bs=64) to 13.0% (bs=256). In order to measure overhead more accurately, all experiments in this section were run on a single GPU. For ResNet-50 and ViT results reported in Table 1, we used a batch size of 1800 (600 per GPU, trained on 3 GPUs), so we estimate the overhead ratio should be even lower. In some cases in Table 8, the Avg. Epoch time of PDT was slightly shorter than baseline, likely due to measurement variance. The total PDT runtime overhead (90–110 ms) is consistent and negligible. The core PDT operations (SVD + prediction + masking) take only $\sim 51$ ms per epoch, which is insignificant compared to the (635s–804s) epoch time.

**3. ViT-Base on ImageNet-1k.** To validate the generality of our overhead analysis across different architectures, we conducted profiling on ViT-Base (Transformer architecture) with ImageNet-1k at a batch size of 256. Table 9 presents detailed memory and runtime measurements for both baseline and PDT.

Despite ViT-Base having significantly more parameters (86.4M) than ResNet-50 (25.6M), the memory overhead breakdown remains consistent with our theoretical analysis: snapshot storage and SVD workspace dominate (40% and 53% respectively), with only 7% attributed to auxiliary memory. The absolute memory overhead (4,954 MB) is larger than ResNet-50 (1,463 MB) as expected from the $\mathcal{O}(N \times h)$ scaling. The runtime overhead remains negligible at 0.11%.

Table 8: Memory and Runtime overhead of PDT compared to baseline across different batch sizes. Trained on ImageNet-1k using ResNet-50, SGD with Momentum baseline optimizer with lr=0.1, momentum=0.9, with CosineAnnealingLR scheduler. All experiments ran on a single Nvidia H100 (80 GB) GPU.

| | Baseline | | | PDT | | |
|---|---|---|---|---|---|---|
| Metric | bs=64 | bs=128 | bs=256 | bs=64 | bs=128 | bs=256 |
| GPU Memory Usage | | | | | | |
|     Peak GPU Memory (MB) | 3,062 | 5,792 | 11,258 | 4,528 | 7,255 | 12,721 |
|     Currently Allocated (MB) | 400 | 436 | 512 | 2,439 | 2,436 | 2,439 |
|     Reserved by Allocator (MB) | 3,404 | 6,766 | 13,424 | 7,392 | 10,864 | 15,298 |
| *PDT Memory Overhead* | | | | | | |
|     Total PDT Overhead (MB) | — | — | — | 1,466 | 1,463 | 1,463 |
|     Overhead Ratio (%) | — | — | — | **47.9%** | **25.3%** | **13.0%** |
| *PDT Memory Overhead Breakdown* | | | | | | |
|     Snapshot Storage (MB) | — | — | — | 585 | 585 | 585 |
|     SVD Workspace (MB) | — | — | — | 780 | 780 | 780 |
|     Other (MB) | — | — | — | 101 | 98 | 98 |
| *PDT Runtime Overhead (per epoch)* | | | | | | |
|     Avg Time per Epoch (s) | 813.9 | 653.1 | 626.5 | 804.0 | 651.5 | 635.3 |
|     Total PDT Runtime Overhead (ms) | — | — | — | 90 | 90 | 110 |
|     Overhead Ratio (%) | — | — | — | **0.011%** | **0.013%** | **0.017%** |
| *PDT Runtime Overhead Breakdown* | | | | | | |
|     **Core PDT Operations (ms)** | — | — | — | 51.5 | 50.7 | 51.1 |
|       ↪ SVD decomposition (ms) | — | — | — | 31.0 | 30.4 | 30.7 |
|       ↪ Prediction (ms) | — | — | — | 19.7 | 19.5 | 19.6 |
|       ↪ Masking (ms) | — | — | — | 0.8 | 0.8 | 0.8 |
|     Auxiliary Operations (ms) | — | — | — | 38.5 | 39.3 | 58.9 |

### A.8.3 SCALABILITY AND GENERALIZATION ANALYSIS

The profiling results align robustly with our theoretical analysis in Appendix A.3, confirming that PDT is scalable to large modern architectures.

AlexNet has a large number of parameters (57M) relative to the small size of the CIFAR-10 dataset. This results in a scenario with high memory requirements for SVD (proportional to $N$) but a very short epoch duration (low compute load). For a large-scale dataset like ImageNet, the computational load per epoch increases dramatically, while the parameter count $N$ (which dictates PDT overhead) does not change with a specific network architecture. The disparity of the memory overhead ratio (AlexNet: 1,145%, ResNet-50: 13%, ViT-Base: 26%) can be explained by the composition of baseline memory. For AlexNet on CIFAR-10, the baseline memory (736 MB) is dominated by model parameters because the input image size ($32 \times 32$) generates very small activation maps. For ImageNet-1k with ($224 \times 224$) inputs, the baseline memory (11–19 GB) is dominated by activation memory, while parameter storage is a smaller fraction. Since PDT overhead scales *only* with parameters and not activations, the relative overhead drops dramatically.

In conclusion, while PDT introduces a linear space complexity $\mathcal{O}(N \times h)$, this overhead is amortized in a large-scale training scenario where memory is dominated by activations and runtime is dominated by gradient computation.

Table 9: Memory and Runtime overhead of PDT compared to baseline. Trained on ImageNet-1k using ViT-Base, batch size=256, AdamW baseline optimizer with lr=0.0006, weight decay=0.05, with CosineAnnealingLR scheduler. All experiments ran on a single Nvidia H100 (80 GB) GPU.

| Metric | Baseline | PDT |
|---|---|---|
| *GPU Memory Usage* | | |
| Peak GPU Memory (MB) | 19,022 | 23,976 |
| Currently Allocated (MB) | 1,536 | 8,406 |
| Reserved by Allocator (MB) | 19,462 | 35,328 |
| *PDT Memory Overhead* | | |
| Total PDT Overhead (MB) | — | 4,954 |
| Overhead Ratio (%) | — | **26.0%** |
| *PDT Memory Overhead Breakdown* | | |
| Snapshot Storage (MB) | — | 1,981 |
| SVD Workspace (MB) | — | 2,642 |
| Other (MB) | — | 330 |
| *PDT Runtime Overhead (per epoch)* | | |
| Avg Time per Epoch (s) | 831.3 | 832.1 |
| Total PDT Runtime Overhead (ms) | — | 877 |
| Overhead Ratio (%) | — | **0.11%** |
| *PDT Runtime Overhead Breakdown* | | |
| **Core PDT Operations (ms)** | — | 193.2 |
| ↪ SVD decomposition (ms) | — | 127.4 |
| ↪ Prediction (ms) | — | 62.8 |
| ↪ Masking (ms) | — | 3.0 |
| Auxiliary Operations (ms) | — | 683.6 |

Table 10: Comprehensive masking ablation results on AlexNet trained on CIFAR-10. Success rates are shown as successful/total runs. Final accuracy reported as mean $\pm$ std across five seeds (successful runs only).

| Configuration | $\tau$ | LR=0.01 | | LR=0.05 | | Overall |
|---|---|---|---|---|---|---|
| | | Success Rate | Final Acc (%) | Success Rate | Final Acc (%) | Success |
| Baseline (SGD) | — | 5/5 | $64.40 \pm 0.71$ | 5/5 | $78.68 \pm 0.21$ | **10/10** |
| Accel Only | 3 | 5/5 | $58.04 \pm 1.29$ | 5/5 | $78.23 \pm 0.64$ | 16/20 |
| | 5 | 4/5 | $48.20 \pm 8.19$ | 2/5 | $10.00 \pm 0.00$ | |
| Consistency Only | 3 | 0/5 | *Crashed* | 1/5 | 10.00 | 1/20 |
| | 5 | 0/5 | *Crashed* | 0/5 | *Crashed* | |
| Full PDT | 3 | 5/5 | $\mathbf{69.39 \pm 0.79}$ | 5/5 | $78.98 \pm 0.18$ | **20/20** |
| | 5 | 5/5 | $69.10 \pm 1.33$ | 5/5 | $\mathbf{79.48 \pm 0.20}$ | |

## A.9  ABLATION STUDY: MASKING CRITERION ANALYSIS

In Sec. 3.2, we introduce a masking mechanism including two principles: the acceleration effectiveness criterion (Eq. 6) and the dynamic consistency criterion (Eq. 7). To investigate the contribution and distinct role of each masking criterion, we conducted a comprehensive ablation study on AlexNet trained on CIFAR-10, using SGD as the base optimizer. We compare four configurations: (1) *Baseline* (standard SGD), (2) *Accel Only* (apply only acceleration effectiveness criterion for masking), (3) *Consistency Only* (apply only dynamic consistency criterion for masking), and (4) *Full PDT* (our complete PDT method with both criteria). Each configuration was evaluated with learning rates 0.01 and 0.05, prediction steps $\tau \in \{3, 5\}$, and five random seeds (0, 100, 200, 300, 400), totaling 70 experimental runs.

Table 10 presents the success rate (proportion of runs that completed without crashing) and final validation accuracy (computed only from successful runs) for each configuration. *Consistency Only*

achieves only 1 successful run (seed=300, LR=0.05, $\tau = 3$) with 10% accuracy (random level), demonstrating catastrophic failure. *Accel Only* achieves 80% success but shows particular instability at $\tau = 5$ with LR=0.05 (2/5 success), with successful runs achieving only 10% accuracy, indicating its inability to predict longer steps. The full PDT achieves perfect robustness (20/20 success) and the highest validation accuracy at both learning rates, outperforming the baseline performance, and validating the necessity of combining both criteria.

Figure 14 visualizes the training dynamics across different learning rates and prediction steps ($\tau$). Since there is no need to adhere to the dynamic consistency criterion, *Accel only* consistently has a higher masked ratio than the full PDT. In all configurations, its train loss is higher than that of full PDT. Specifically, in Fig. 14(d) (LR=0.05,$\tau = 5$), the *Accel Only* validation accuracy collapses to random chance (10%) after the initial epochs. Without the consistency criterion, too many predictions with incorrect directions are accepted, which causes gradient explosion, especially when predicting multiple steps. In contrast, *Full PDT* consistently achieves the lowest training loss and highest validation accuracy across all configurations. The shaded variance regions for *Full PDT* are notably tighter than those for *Accel Only*, demonstrating that combining both criteria makes the training process significantly more stable.

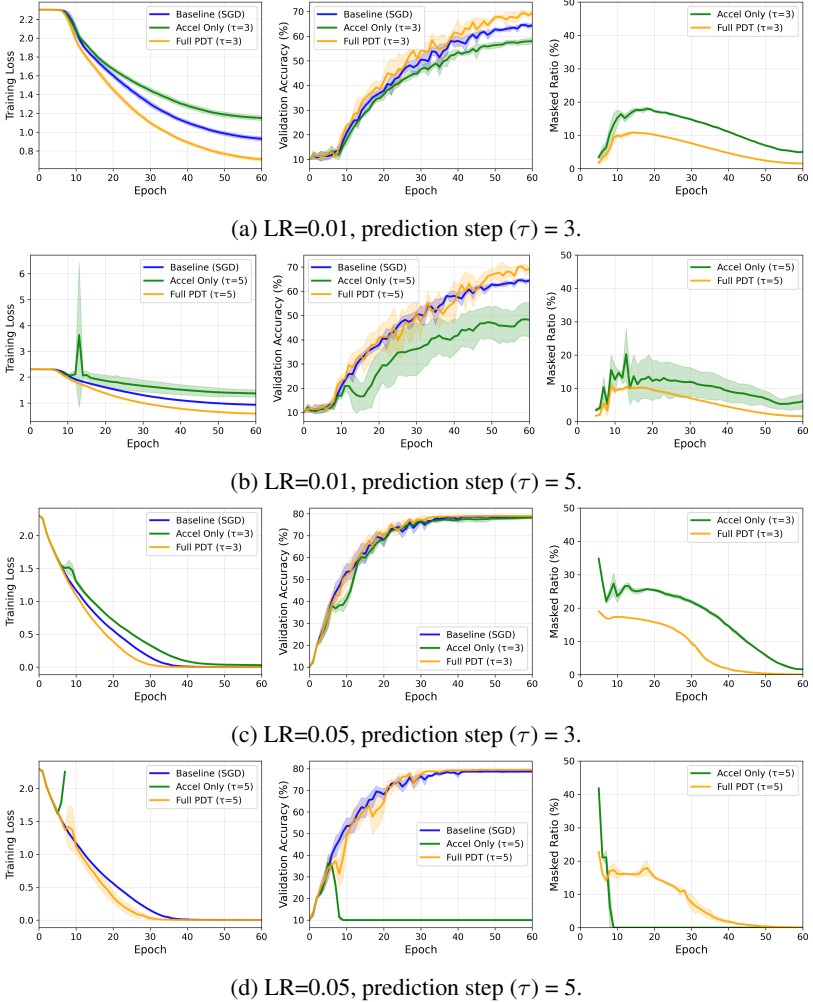

(a) LR=0.01, prediction step ($\tau$) = 3.

(b) LR=0.01, prediction step ($\tau$) = 5.

(c) LR=0.05, prediction step ($\tau$) = 3.

(d) LR=0.05, prediction step ($\tau$) = 5.

Figure 14: Mean training curves (with standard deviation indicated by Shaded regions) and performance comparison of baseline and pdt with different masking criteria. Trained on CIFAR-10 using AlexNet, batch size = 256, with CosineAnnealingLR scheduler. Only successful runs are included.

To further investigate the contribution of each masking criterion across different training stages, we decompose the predictions rejected by PDT and examine which criterion rejects them. Figure 15

shows the temporal evolution of each criterion's contribution throughout training. We analyze the *Full PDT* configuration (LR=0.05) across five random seeds to understand how the acceptance and rejection ratios evolve. Experimental results from $\tau$=3 and $\tau$=5 show a similar pattern. Ratio of predictions rejected by Consistency Criterion (Eq. 7) due to opposite direction remains stable at $\sim$50% throughout training. Ratio of predictions rejected by Acceleration Criterion Lower Bound (Eq. 6) (for being too small ($< 1\times$ SGD update)) decreases to near zero, indicating this bound is primarily active early in training. Ratio of predictions rejected by Accel Upper Bound (for being too large ($> (\tau)\times$ SGD update)) increases dramatically (from $\sim$20% to $\sim$50%), becoming critical in the later training stage.

The overall mask ratio decline is due to the increasing rejection by the upper bound of the acceleration effectiveness criterion. The gradual decline in mask ratio reflects the changing optimization landscape: early in training, when the loss landscape is steep and gradients are large, DMD sometimes underestimates the required step size. However, as training progresses and gradients shrink near convergence, DMD predictions rarely fall below the minimum threshold. When near convergence, gradients become small, noisy, and oscillatory. The upper bound becomes critical for preventing divergence by rejecting these over-aggressive predictions.

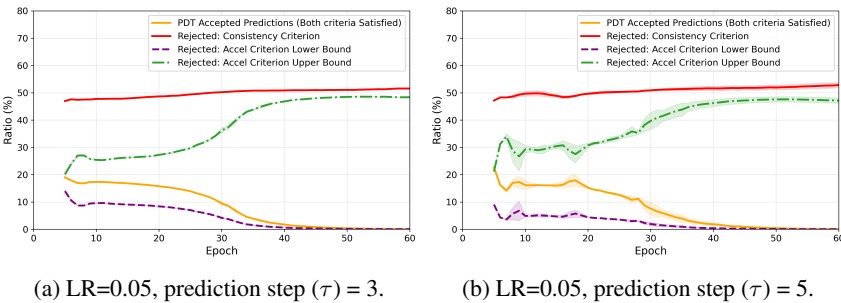

(a) LR=0.05, prediction step ($\tau$) = 3.  (b) LR=0.05, prediction step ($\tau$) = 5.

Figure 15: Temporal evolution of masking criteria contributions during training. Results from full PDT (both criteria applied). Trained on CIFAR-10 using AlexNet, LR=0.05, batch size = 256, with CosineAnnealingLR scheduler. Only successful runs are included.

From the above ablation results, compared to a single criterion, the complete PDT with both criteria has smooth loss curves, faster convergence, and low variance across seeds. This provides strong empirical evidence supporting our theoretical design and demonstrates that neither criterion alone is sufficient. Both criteria play distinct, complementary roles. The dynamic consistency criterion provides stable, stage-agnostic directional filtering. The acceleration effectiveness criterion ensures predictions provide speedup by enforcing magnitude bounds. During the early training stage, the lower bound guarantees that each accepted prediction moves the weights at least as far as a single SGD step, enabling acceleration. The upper bound prevents over-aggressive updates that could cause divergence and ensures stability throughout the training process.

## A.10    IMPACT OF DIFFERENT INITIAL LEARNING RATES ON MASKED RATIO

To investigate the impact of different initial learning rates on masked ratio behavior and computational efficiency, we conducted systematic experiments on AlexNet trained on CIFAR-10. We compare Baseline (SGD) and PDT (prediction steps $\tau = 5$, prediction interval $T_i = 1$, past snapshot counts $h = 5$, starting epoch $T_0 = 5$) across four different learning rates: 0.001, 0.01, 0.05, and 0.1. All experiments use batch size 256, CosineAnnealingLR scheduler with $\text{lr}_{\min} = 10^{-3}$, and train for 60 epochs. Each configuration was evaluated with three random seeds (0, 100, 200). We measure mask acceptance ratio by training stage (early: epochs 5–20, mid: epochs 21–40, late: epochs 41–60) and final validation accuracy.

As shown in Table 11, larger learning rates lead to higher mask acceptance ratios during early training. Specifically, LR=0.1 achieves the highest early-training masked ratio (18.77%), followed by LR=0.05 (16.94%), LR=0.01 (8.12%), and LR=0.001 (0.06%). For relatively large learning rates (e.g., 0.05 and 0.1), this trend is not strictly observed. As training progresses, the masked ratios across different learning rates decline to varying degrees. However, for extremely small learning rates (e.g., 0.001),

the masked ratio remains consistently low. PDT consistently improves over baseline across all learning rates, demonstrating robustness.

Table 11: Mask acceptance ratio and validation accuracy across learning rates: mask acceptance ratio by training stage and final test accuracy (mean $\pm$ std across 3 seeds). Training stages: Early (epochs 5–20), Mid (epochs 21–40), Late (epochs 41–60).

| LR | Early (%) | Mid (%) | Late (%) | Overall (%) | Base Acc (%) | PDT Acc (%) |
|---|---|---|---|---|---|---|
| 0.001 | $0.06 \pm 0.00$ | $0.07 \pm 0.00$ | $0.10 \pm 0.02$ | $0.08 \pm 0.01$ | $11.83 \pm 1.04$ | $14.92 \pm 2.44$ |
| 0.01 | $8.12 \pm 0.20$ | $7.10 \pm 0.06$ | $\mathbf{2.77 \pm 0.02}$ | $5.77 \pm 0.07$ | $63.96 \pm 0.14$ | $68.43 \pm 0.64$ |
| 0.05 | $16.94 \pm 0.20$ | $\mathbf{8.52 \pm 0.5}$ | $0.68 \pm 0.15$ | $\mathbf{7.94 \pm 0.31}$ | $78.67 \pm 0.13$ | $\mathbf{79.24 \pm 0.22}$ |
| 0.1 | $\mathbf{18.77 \pm 0.24}$ | $7.10 \pm 0.52$ | $0.37 \pm 0.05$ | $7.91 \pm 0.20$ | $79.76 \pm 0.27$ | $\mathbf{80.06 \pm 0.20}$ |

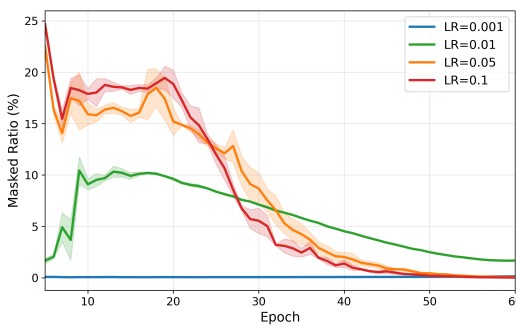

Figure 16: Mask acceptance ratio evolution across different learning rates (mean $\pm$ std shading across 3 seeds). Trained on CIFAR-10 using AlexNet, batch size = 256, with CosineAnnealingLR scheduler.

Figure 16 visualizes the mask ratio evolution across different learning rates. Early training exhibits the highest acceptance, as the loss landscape is steep and gradients provide a strong signal. By late training (epochs 41–60), mask ratios converge to low values across all learning rates.

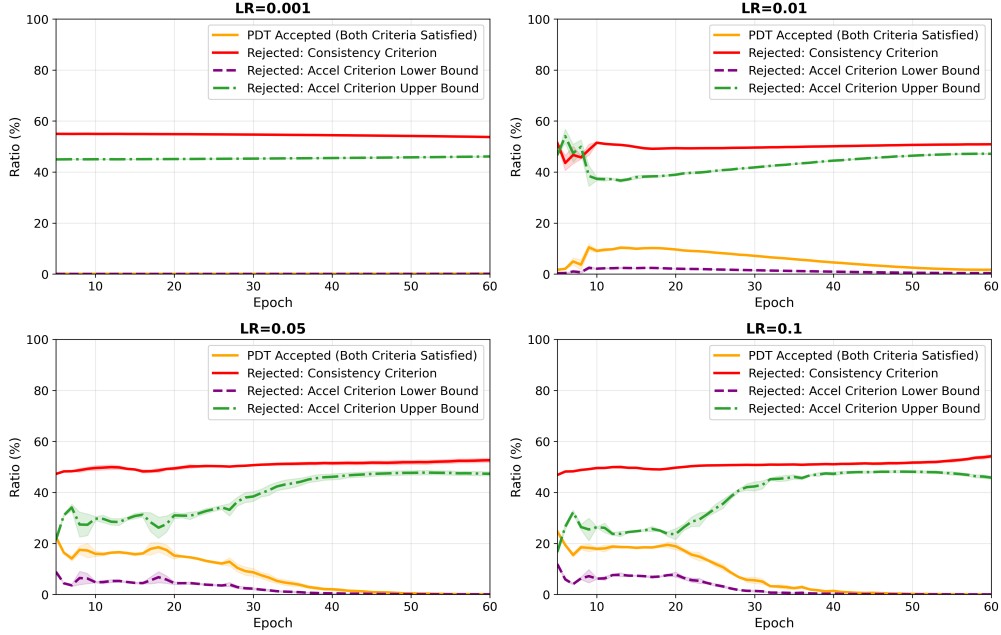

Figure 17: Temporal evolution of masking criteria contribution across learning rates. Trained on CIFAR-10 using AlexNet, batch size = 256, with CosineAnnealingLR scheduler.

The temporal evolution of masking criteria contribution shown in Fig. 17 reveals why extremely small learning rates (e.g., 0.001) lead to consistently low mask acceptance ratios during the entire training stage. The mask acceptance behavior is governed by the interaction between the learning rate and the Acceleration Effectiveness Criterion (Eq. 6). At LR=0.001, the figure shows that predictions are overwhelmingly rejected by the consistency criterion and the *Upper Bound* of acceleration criterion (Green line). When the learning rate is extremely low (e.g., 0.001), the gradient steps are small, making the "allowable acceleration window" microscopic. Although DMD predicts a trajectory based on historical dynamics, the magnitude of this prediction, even if small, easily exceeds the excessively strict upper bound imposed by the tiny learning rate. In contrast, at LR=0.05 and 0.1, the upper bound constraint is relaxed. The gradient signal is strong enough that the SGD step size is comparable to the DMD prediction scale. Consequently, a larger proportion of predictions fall comfortably *between* the lower bound and the upper bound, resulting in a significantly higher masked ratio.

In conclusion, the stage-wise analysis reveals that masked ratios are highest during early training and decline adaptively toward convergence, prioritizing stability over acceleration in the final stages. Contrary to the intuition that smaller steps imply stability and higher acceptance, our analysis confirms that larger learning rates are necessary to generate the clear dynamic signals required for high-fidelity DMD predictions. PDT is most effective when the baseline optimizer takes steps large enough to define a "permissible region" that accommodates the scale of DMD's spectral predictions.

### A.11 EFFECT OF NON-I.I.D. TRAINING DATA

We further investigate the robustness of PDT under some challenging training conditions. For example, when the batch is too small for a diverse dataset like ImageNet, the weight updates could be chaotic since each consecutive batch is no longer an identical distribution. There are two experimental designs that can test this: 1) test PDT on a very large dataset like ImageNet-22K and 2) design a batching scheme to intentionally violate the i.i.d. assumption of mini-batches using a smaller dataset such as CIFAR-10. In the second design, we maintain the normal batch size, but only put samples of the same class in the batch. We also randomize the batch sequence instead of using any fixed order so that there is no regular training set dynamics that DMD might pick up on.

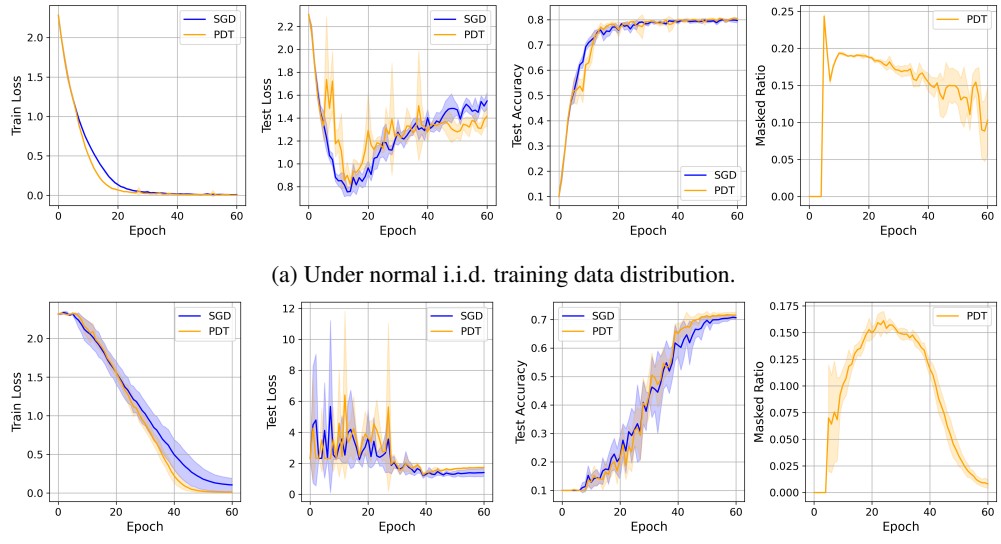

(a) Under normal i.i.d. training data distribution.

(b) Under non-i.i.d. training data distribution.

Figure 18: Performance comparison between SGD and PDT under i.i.d. and non-i.i.d. training data distributions, with the same hyperparameters configuration. Trained on CIFAR-10 using AlexNet, batch size=128, lr=0.05, with CosineAnnealingLR scheduler. The shaded areas represent the standard deviation across 5 runs with different random seeds (0, 100, 200, 300, 400).

Fig. 18 and Table 12 show the performance and runtime comparison between SGD and PDT under the non-i.i.d. setting using the second experimental design since non-i.i.d. is guaranteed. We preserve

Table 12: Performance and runtime comparison between SGD and PDT under i.i.d. and non-i.i.d. training data distributions, with the same hyperparameter configuration. Trained on CIFAR-10 using AlexNet, batch size=128, lr=0.05, with CosineAnnealingLR scheduler.

| Training Data Distribution | Method | Final Accuracy (mean $\pm$ std) | Best Train Loss (mean $\pm$ std) | Time to Baseline Best Loss (s) (mean $\pm$ std) | Runtime Reduction (%) |
|---|---|---|---|---|---|
| i.i.d. | SGD | $0.7969 \pm 0.0093$ | $0.0039 \pm 0.0017$ | $662.48 \pm 7.73$ | 9.15 |
| | PDT | $0.8011 \pm 0.0067$ | $0.0016 \pm 0.0017$ | $601.86 \pm 17.78$ | |
| non-i.i.d. | SGD | $0.7067 \pm 0.0062$ | $0.1053 \pm 0.0874$ | $806.83 \pm 13.15$ | 27.90 |
| | PDT | $0.7159 \pm 0.0103$ | $0.0119 \pm 0.0057$ | $581.73 \pm 19.34$ | |

the original i.i.d. sampling of the validation set. All experiments are repeated with five random seeds (0, 100, 200, 300, 400) to ensure statistical significance.

We make some interesting observations. First, despite the challenging non-i.i.d. setup, PDT still achieves better performance than SGD in terms of faster convergence without sacrificing accuracy. However, we also observe that in the non-i.i.d. case, learning starts out much more slowly for both SGD and PDT and both take longer to converge. Second, in the non-i.i.d. case, the variance of each of the performance curves is generally larger than those of the i.i.d. case. This is because the model needs to handle more abrupt transitions between different class distributions.

Fig. 18 and Table 12 further demonstrate that PDT's advantage extends beyond standard i.i.d. training conditions, showing its robustness to challenging data sets where traditional assumptions about data distribution are violated.

## A.12 Cross-Domain Evaluation on Natural Language Processing

To demonstrate the generalization of PDT beyond computer vision, we evaluate its effectiveness on text classification tasks. This experiment validates PDT's applicability to a fundamentally different data modality (discrete text) and architecture (Recurrent Neural Networks).

We employ a deep LSTM network for 4-class topic classification on the AG News dataset (Zhang et al., 2015; Gulli). The architecture consists of an embedding layer, a 4-layer stacked LSTM (512 hidden units per layer), and a linear classifier, totaling 8.25M parameters. This setup differs significantly from CNNs, particularly in terms of gradient flow dynamics (backpropagation through time). We use SGD with a learning rate of 0.1 and batch size 128 for 30 epochs. The PDT hyperparameters are identical to those used in our vision experiments ($\tau = 5, h = 5, T_0 = 5, T_i = 1$) to assess the robustness of PDT without domain-specific tuning. Results are averaged over 3 random seeds.

Table 13: Performance comparison on AG News dataset (text classification) using a deep LSTM model. Trained on a single Nvidia RTX A6000 GPU for 30 epochs, lr=0.1, batch size=128, with CosineAnnealingLR scheduler.

| Optimizer | Final Accuracy | TTB-Loss (s) | TTB-Acc (s) | Runtime Reduction (%) | |
|---|---|---|---|---|---|
| | | | | Train Loss | Val. Acc. |
| SGD | $86.10 \pm 1.30$ | $700.8 \pm 72.1$ | $694.4 \pm 76.9$ | **26.1** | **20.5** |
| PDT | $88.44 \pm 0.59$ | $517.9 \pm 102.2$ | $551.7 \pm 120.5$ | | |

As summarized in Table 13, PDT successfully accelerates training in the NLP domain. PDT achieves the baseline's best training loss 26.1% faster and best validation accuracy 20.5% faster. This speedup magnitude is comparable to that observed in our vision experiments (10–40%). PDT also achieves a higher validation accuracy of (88.44% compared to the baseline's 86.10%). Fig. 19 presents the training dynamics comparison between baseline SGD and PDT on the AG News dataset.

The successful application of PDT to text classification with LSTMs provides strong evidence for its cross-domain generalizability. Despite the different optimization landscape of LSTMs, PDT achieves consistent speedups across both CNN-based vision tasks and RNN-based language tasks.

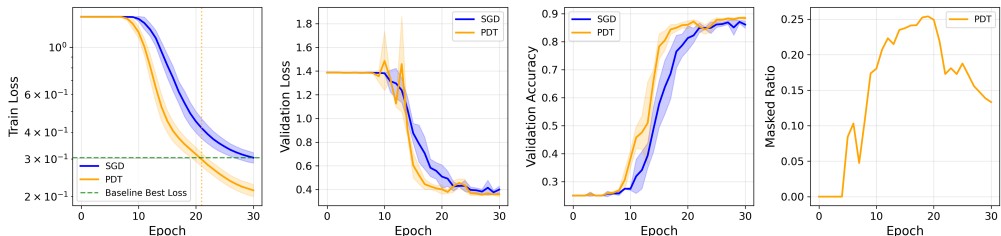

Figure 19: Performance comparison between baseline optimizer and PDT. Trained on the AG News dataset using an LSTM model. Trained on a single Nvidia RTX A6000 GPU for 30 epochs, lr=0.1, batch size=128, with CosineAnnealingLR scheduler. The shaded areas represent the standard deviation across three runs with different random seeds (0, 100, 200).

