# OpenReview forum: "Predictive Differential Training Guided by Training Dynamics"
_ICLR.cc/2026/Conference — ICLR 2026 Poster_

### Official Review · Reviewer_JHov · 2025-10-28

**Soundness:** 3
**Presentation:** 3
**Contribution:** 3
**Rating:** 4
**Confidence:** 4

**Summary:**

*disclaimer* I only used LLM for understanding of part of sec 3.2.

The paper proposes an acceleration strategy for accelerating first order methods (SGD and variants) esp. in large scale deep learning systems. The strategy is to use linear approximation to approximate the nonlinear dynamics of weight updates in DNN with first order methods. In particular, the authors propose a predictive differential training  which serves as plug in tool in SGD or variant optimizer. It starts with SGD updates for a few steps and then use these snapshots to linear approximate and forecast a few steps in the future.  To accept the forecasted weights one by one, the author proposed the two criteria: acceleration effectiveness criterion and dynamic consistency criterion so that the predicted dynamics can be in accordance to the global optimizer dynamics. Empirical results show the effective acceleration on large scale systems in Vision applications and networks.

**Strengths:**

Strengths include 1) theoretical inspired by koopman operator theory . The authors adapts to a computational efficient approach using linear approximation. (and some numerical decomposition for predicted weights) 2) the authors also show the justification of using masking strategy to make “Acceleration” possible and meaningful. 3). The quality of experiments are very high by not only showing the loss and acc reduction but also computation analysis, and show a variety of 1st order optimizers . 3) the topic is interesting in DL optimization community. 4) the presention of this paper is always good to follow.

**Weaknesses:**

Weakness include 1).  No thorough literature search in this line of work: for example a) Introspection:accelerating neural network training by learning weight evolution. b) Learning to Boost Training by Periodic Nowcasting Near Future Weights and a few other following this line of work.  Correspondingly, authors should include some baselines from this line of work for highlighting the superiority of proposed method  2) The proof of convergence should be discussed in more detail for theorem 1.

**Questions:**

1 In Line 371, the paper states: “the masked ratio always starts with higher values in the early stage of the training process, then generally decreases as training progresses.” The authors further suggest that this reflects the increasing complexity of training dynamics in large networks on large datasets, making them harder to predict. I am not entirely convinced by this explanation.
If I understand correctly, the weight is masked  means the predicted weight is accept.
Intuitively, this trend may also be influenced by how stochastic gradient descent (and its variants) behaves: early in training, the loss landscape is typically easier to optimize, leading to faster reduction in loss and more stable gradient directions, which in turn may allow more weights to pass the masking criteria. Later in training, as the optimizer approaches (local) minima, gradients oscillate more around the optimum, making it harder to accept predicted weights for prediction horizons like 5 steps. Could the authors clarify or comment on whether this trend may be more related to optimizer behavior rather than intrinsic “complexity of the dynamics”?

2. Has the authors conducted experiments analyzing the impact of different initial learning rates on the masked ratio behavior? Intuitively, a smaller learning rate might lead to more stable gradient directions, thereby increasing the proportion of weights that satisfy the masking criteria early in training. Conversely, larger initial learning rates might introduce more variability and lower the acceptance ratio. That also affects the computational analysis and efficiency of the current approach in the appendix.

3. The current empirical evaluation focuses exclusively on vision tasks (e.g., AlexNet, ResNet, ViT). While these results are compelling, it would significantly strengthen the work if the method were also evaluated on other domains, such as NLP or speech models. This would help demonstrate the generality and robustness of the PDT framework across architectures and modalities.

---

> ### Author Response · Authors · 2025-11-24
> **Response to Reviewer JHov (part1)**
>
> We sincerely thank the reviewer for the insightful comments. We will address your concerns one by one in the following.
>
> **Weakness 1: More literature search in this line of work.**
>
> We appreciate the reviewer’s pointer to the line of work on predicting future weights, including Introspection, WNN, and NiNo. These methods rely on training separate neural predictors—either per-weight regressors or graph-structured GNNs—using curated checkpoint datasets. Their performance depends on the composition of these meta-training datasets, and adapting them to new architectures or tasks requires rebuilding or retraining the predictor, which makes establishing a universal baseline non-trivial. Moreover, existing works mainly report reductions in training steps and do not provide detailed breakdowns of the inference overhead when the predictor must be applied to each weight or edge, a cost that scales with model size and can involve significant compute or memory. In contrast, PDT requires no meta-training and performs prediction through an analytic Koopman/DMD operator built online from a small number of recent snapshots, without per-weight neural inference. We expanded the related-work section to clearly position PDT relative to these approaches. Please see highlighted toward the end of Sec. 2 in the revised paper.
>
> Separately, we note that the submitted version already compares PDT with two existing DMD-based weight-prediction baselines (Figs. 2 and 6).  Both have been demonstrated only on small-scale models rather than the large architectures as studied in our experiments.
>
> **Weakness 2: The proof of convergence.**
>
> Regarding the convergence analysis in Appendix A.5, we intended to provide a high-level sketch explaining why the masking mechanism prevents divergence when predicted updates deviate from the local optimization direction. We agree that the current presentation can be made more explicit. In the revised paper, **we have expanded the convergence proof in Appendix A.5**. We have included a complete derivation based on standard L-smoothness assumptions, together with the two criteria from Sec. 3.2, showing how these yield a GD-style descent inequality under the prescribed update rule.
>
> **Question 1: The explanation of masked ratio evolution.**
>
> This is an extremely insightful observation! We actually made two interesting observations. One is what’s in the original text, that “the masked ratio starts higher in the early stage of training, then decreases as training progresses”. The second is what we commented out to save space, that “More interestingly, we observe that smaller networks on simpler tasks (FCN/AlexNet on CIFAR-10) show a relatively more gradual reduction in the masked ratio, while larger networks on more complex tasks (ResNet-50/ViT on ImageNet) exhibit a much sharper reduction of masked ratio, especially at the early stage of the training process.” We have added this second observation back (highlighted) now that we have more space. The explanation about “increasing complexity of training dynamics in large networks on large datasets making them more difficult to predict” refers to the second observation. The reviewer’s insight, however, perfectly explains the first observation. We have added the reviewer’s hypothesis to the paper (see highlighted).
>
> We thank the reviewer’s insightful observation. It helped us recognize and more clearly express the key factor driving the masked ratio curve. Please also see a more detailed discussion in Appendix A.7 where we dived deeper into the trend of masked ratio with respect to different layers of the network. We found that the convolutional layers consistently maintain a higher masked ratio as compared to the fully connected layers throughout the training process. Moreover, we have conducted a comprehensive ablation study for masking criterion, and added a detailed analysis in Appendix A.11 in the revised paper.

---

> ### Author Response · Authors · 2025-11-24
> **Response to Reviewer JHov (part2)**
>
> **Question 2: Regarding the impact of different initial learning rates on the masked ratio behavior.**
>
> We thank the reviewer for this insightful hypothesis. It is indeed intuitive to think that smaller learning rates would lead to more stable gradients and thus a higher mask acceptance ratio. However, our new experiments reveal the opposite trend: larger learning rates consistently lead to higher mask acceptance ratios, especially in the early stage of training.
>
> To investigate the impact of different initial learning rates on masked ratio behavior, we have conducted a systematic study on AlexNet trained on CIFAR-10 with learning rates {0.001, 0.01, 0.05, 0.1} and added the detailed analysis in Appendix A.12 (Table 12 and Figs. 17-18).
>
> **Table 12: Mask acceptance ratio and validation accuracy across learning rates. Training stages: Early (epochs 5--20), Mid (epochs 21--40), Late (epochs 41--60).**
> | LR | Early (%) | Mid (%) | Late (%) | Overall (%) | Base Acc (%) | PDT Acc (%) |
> | :--- | :---: | :---: | :---: | :---: | :---: | :---: |
> | 0.001 | 0.06 ± 0.00 | 0.07 ± 0.00 | 0.10 ± 0.02 | 0.08 ± 0.01 | 11.83 ± 1.04 | 14.92 ± 2.44 |
> | 0.01 | 8.12 ± 0.20 | 7.10 ± 0.06 | **2.77 ± 0.02** | 5.77 ± 0.07 | 63.96 ± 0.14 | 68.43 ± 0.64 |
> | 0.05 | 16.94 ± 0.20 | **8.52 ± 0.5** | 0.68 ± 0.15 | **7.94 ± 0.31** | 78.67 ± 0.13 | 79.24 ± 0.22 |
> | 0.1 | **18.77 ± 0.24** | 7.10 ± 0.52 | 0.37 ± 0.05 | 7.91 ± 0.20 | 79.76 ± 0.27 | **80.06 ± 0.20** |
>
> As shown in Table 12, larger learning rates lead to higher mask acceptance ratios during early training. Specifically, LR=0.1 achieves the highest early-training masked ratio (18.77\%), followed by LR=0.05 (16.94\%), LR=0.01 (8.12\%), and LR=0.001 (0.06\%). For relatively large learning rates (e.g., 0.05 and 0.1), this trend is not strictly observed. As training progresses, the masked ratios across different learning rates decline to varying degrees. However, for extremely small learning rates (e.g., 0.001), the masked ratio remains consistently low.
>
> Our component-wise analysis (Fig. 18 in Appendix A.12) reveals why extremely small learning rates (e.g., 0.001) lead to consistently low mask acceptance ratios during the entire training stage. The bottleneck is not just directional stability, but the Acceleration Effectiveness Criterion (Eq. 6), specifically the Upper Bound:
>
>  $\| \mathbf{w}_{i+1}^{\mathrm{opt}} - \mathbf{w}_i^{\mathrm{opt}} \| < \| \mathbf{w}\_{i+\tau}^{\mathrm{pred}} - \mathbf{w}\_i^{\mathrm{opt}} \| \le \tau \| \mathbf{w}\_{i+1}^{\mathrm{opt}} - \mathbf{w}\_i^{\mathrm{opt}} \|$
>
> The upper bound scales linearly with the learning rate $\eta$. At LR = 0.001, the gradient steps are small, and the "allowable acceleration window" becomes microscopic. The Fig.18 shows that predictions are overwhelmingly rejected by the consistency criterion (Eq.7) and the Upper Bound of acceleration criterion (Eq.6). Even if DMD predicts a directionally correct trajectory, its magnitude almost always exceeds this excessively strict limit imposed by the tiny SGD step. In contrast, at Large LR (e.g., 0.1), the gradient signal is strong, and the permissible window is large enough to accommodate the scale of DMD predictions. Consequently, a larger proportion of predictions fall comfortably between the lower bound and the upper bound, resulting in a significantly higher masked ratio.
>
> We have incorporated this analysis into Appendix A.12  to clarify the interaction between learning rates and our masking mechanism. We have also conducted a comprehensive ablation study for masking criterion, and added a detailed analysis in Appendix A.11.
>
> In addition, about computational analysis and efficiency of PDT, we have added a detailed "Profiling and Overhead Analysis" section in Appendix A.10. Our profiling on AlexNet with CIFAR10 (Table 8), ResNet-50 and Vit-base on ImageNet-1k (Table 9-10) show that the PDT runtime overhead is negligible (0.017% for ResNet-50, and 0.11% for ViT-base).
>
> **Question 3: Evaluate PDT on other domains such as NLP.**
>
> Thanks for the suggestion! It would indeed be interesting to test PDT on application domains other than vision. We are working on the experiments at the moment. Hopefully, we’ll have some results before the end of the discussion phase.

---

> ### Author Response · Authors · 2025-12-04
> **Update on Question 3: the cross-domain experiments**
>
> **Update on Question 3: the cross-domain experiments**
>
> We thank the reviewer for this insightful suggestion. We have conducted additional experiments on text classification using RNNs to validate PDT's cross-domain generalizability, and have added the detailed results to **"Appendix A.13 Cross-Domain Evaluation on Natural Language Processing**".
>
> We trained a deep LSTM network (4 layers, 8.25M parameters) from scratch for 4-class topic classification on the *AG News* dataset. This setup challenges PDT with a fundamentally different data modality (discrete text) and architecture (RNNs), which exhibit distinct gradient dynamics compared to CNNs/ViTs. We use SGD with a learning rate of 0.1 and a batch size of 128. The PDT hyperparameters are identical to those used in our vision experiments ($\tau=5, h=5, T_0=5, T_i=1$) to assess the robustness of PDT without domain-specific tuning.
>
> **Table 13: Performance comparison between baseline optimizer and PDT. Trained on the AG News dataset using an LSTM model.**
> | Optimizer | Final Accuracy | TTB-Loss (s) | TTB-Acc (s) | Runtime Red. (Loss %) | Runtime Red. (Acc. %) |
> | :--- | :---: | :---: | :---: | :---: | :---: |
> | SGD | 86.10 ± 1.30 | 700.8 ± 72.1 | 694.4 ± 76.9 | — | — |
> | PDT | 88.44 ± 0.59 | 517.9 ± 102.2 | 551.7 ± 120.5 | **26.1** | **20.5** |
>
> As shown in Table 13, PDT successfully generalizes to the NLP domain. PDT achieved the baseline's best training loss **26.1% faster** and best validation accuracy **20.5% faster**. The magnitude of the speedup (~26%) is consistent with our vision experiments (10-40%). This confirms that PDT's core principle is robust and effective for both feedforward (CNN/ViT) and recurrent (LSTM) architectures.

---

### Official Review · Reviewer_SwkT · 2025-10-31

**Soundness:** 4
**Presentation:** 4
**Contribution:** 4
**Rating:** 8
**Confidence:** 4

**Summary:**

This paper addresses the problem of accelerating neural network training through predictive differential training - that is, predicting the future weight values of the network during training to bypass gradient descent steps. To predict the weight values at future epochs, the authors apply dynamic mode decomposition (DMD) over a small window of past weight values. The authors then show that DMD alone is insufficient for stable training, and propose masking strategy that determines which parameters will be accelerated via DMD, and which will use SGD-based updates. Through extensive experiments, the authors demonstrate that their algorithm can enhance many different types of optimizers. Also the proposed algorithm is found to work on both image classification and self-supervised learning, further highlighting the robustness of the study.

**Strengths:**

1.	The proposed algorithm has multiple well motivated components, that ultimately lead to significant speed ups across multiple architectures and learning tasks. The study is not a simple application of DMD either, as the authors demonstrate that their masking strategy is crucial in speeding up model training.
2.	The proposed method seems to be quite robust, showing performance improvement across not just different architectures, but also completely different problem types (self-supervised learning), optimizers, and out-of-distribution minibatch situations (A.8). These results are quite impressive.
3.	The paper is very clear, with the motivation and details of the algorithm clearly laid out. The use of toy problem also helps intuition and the authors provide extensive information about their experiments in the appendix, making their paper quite transparent.

**Weaknesses:**

1.	The convergence analysis in A.5 seems only to be a sketch of a proof and not a proper convergence proof. Furthermore, the inequality direction in equation 8 is different from the one show in the main text (equation 6).
2.	The proposed PDT method introduces 4 additional hyperparameters, which is a bit too many to select via brute force hyperparameter search. A heuristic for determining these values will be handy.

**Questions:**

1.	While the authors have adequately provided information on the computational complexity of their algorithm, I am also curious about the memory complexity. Naively thinking, I imagine that the additional memory required would scale as $N*h$ and then some more, related to the cost of performing SVD on a $N\times h$ matrix. Can the authors provide a comparison between their algorithm and SGD on this front? Does this additional memory requirement present any difficulties, or is this not an issue in practice?

2. For the proposed masking strategy, I am curious about the relative importance between the acceleration effectiveness criterion and the dynamic consistency criterion. Can the authors present an ablation experiment by using only one or the other? I am also curious if the changes in the masking ratio during training (A.7) is related to the different contributions of the two masking criteria during different stages of training.

3. The authors do mention that their method comes with couple of additional parameters and discusses their effect in A.9. Regarding this point, can the authors provide a rule of thumb for selecting these values given a training problem?

4. I am confused about the direction of the inequality in the acceleration criterion. While the main text seems so suggest that equation (6) is correct, the direction of the inequality between equation (6) and (8) is different. Can the authors clarify?

5. For each of the experiments, can the authors provide information of how many batches were in an epoch? I wan to get a better idea of how many gradient steps are being skipped by PDT method.

---

> ### Author Response · Authors · 2025-11-24
> **Response to Reviewer SwkT (part1)**
>
> We sincerely thank the reviewer for the positive assessment of our work and the insightful comments. We will address your concerns one by one in the following.
>
> **Weakness 1: The mismatch between Eq.6 and Eq.8, and the convergence proof.**
>
> We thank the reviewer for raising this critical point. The mismatch between Eq. 6 in the main text and Eq. 8 in the appendix is indeed a typo during paper editing. The original acceleration effectiveness criterion has a lower bound and an upper bound . The lower bound ensures that the prediction-based weight update is at least larger than the one-step optimization-based weight update (i.e., $||w_{i+\tau}^{pred} - w_i^{opt}|| > ||w_{i+1}^{opt} - w_i^{opt}||$). The upper bound is set as $\tau$ times the one-step optimization-based weight update to ensure that the prediction-based weight update is not excessively large, reducing the risk of divergence. (I.e., $||w_{i+\tau}^{pred} - w_i^{opt}|| \le \tau ||w_{i+1}^{opt} - w_i^{opt}||)$. Here, $\tau$ is the prediction step.  While we were trying to fit the paper within the page limit, some excessive edits resulted in the inconsistency.
>
> We have corrected Eq.6 in Section 3.2 (highlighted), with a more succinct explanation.
>
> Regarding the convergence analysis in Appendix A.5, we intended to provide a high-level sketch explaining why the masking mechanism prevents divergence when predicted updates deviate from the local optimization direction. We agree that the current presentation can be made more explicit. In the revised paper, **we have expanded the convergence proof in Appendix A.5**. We have included a complete derivation based on standard L-smoothness assumptions, together with the two criteria from Sec. 3.2, showing how these yield a GD-style descent inequality under the prescribed update rule.
>
> **Weakness 2: Four additional hyperparameters introduced by PDT.**
>
> We totally agree that four hyperparameters, especially the potential number of combinations, can be a bit too many, and practical guidelines are valuable for adoption. In this paper, these four hyperparameters are always set to prediction step=5, prediction interval=1, start epoch=5, and past snapshot counts=5. Overall, these values are robust across optimizers, architectures, and datasets. See a detailed study about the effect of these four PDT-related hyperparameters in Appendix A.9 (Figure 14).  Based on these studies, we propose the following "Rule of Thumb" guidelines to help find the appropriate hyperparameters for different scenarios.
>
> - Past Snapshot counts ($h$): $h=5$ is a “sweet spot”. Smaller $h$ is insufficient for capturing dynamics, larger $h$ includes “stale” weights from much earlier training and introduces additional overhead. For networks with extremely dynamic changes, a larger value of $h$ (e.g., $h$=10) is also worth trying.
> - Prediction Steps ($\tau$): We recommend starting with $\tau=5$ as a robust default value. This value provides a good balance between acceleration benefit and prediction accuracy across diverse architectures and datasets. Users can increase $\tau$ to 7 if their training loss curves are very stable and exhibit minimal variance. Conversely, if gradient explosion or instability occurs, reducing $\tau$ to 3 provides a more conservative acceleration while maintaining stability. A practical configuration is to set $\tau \in [3, 7]$. Too large leads to divergence, while too small makes it meaningless.
> - Prediction Interval ($T_i$): We recommend setting $T_i=1$ as the default. If the training process is unstable, then gradually increase the interval.
> - Start Epoch ($T_0$): The start epoch $T_0$ should typically be equal to $h$ to ensure sufficient history is available for the first prediction.
>
> We have added this heuristic guide to Appendix A.9. See highlighted.

---

> ### Author Response · Authors · 2025-11-24
> **Response to Reviewer SwkT (part2)**
>
> **Question 1: The memory complexity of PDT.**
>
> We agree that memory complexity is an integrative part of the complexity analysis. The memory complexity of PDT comes primarily from storing the past $h$ parameter snapshots used to form the weight trajectory matrix for the Koopman/DMD approximation, which scales as $\mathcal{O}(N \times h)$, where $N$ denotes the number of model parameters and $h$ denotes the number of past epochs retained. In all experiments, we intentionally keep $h$ small (set to 5), which limits the additional memory overhead. These snapshots are stored only temporarily and are overwritten after each DMD computation, so the space cost does not accumulate over training. The successful experiments on ViT-Huge (632M parameters) showcased the feasibility of the additional memory requirement from DMD.
>
> We have added the above space complexity analysis in Appendix A.4 (see highlighted). We have also added detailed memory profiling tables in Appendix A.10 showing peak GPU memory usage, PDT memory overhead, and detailed overhead for each PDT component. (Please see Tables 8-10 below and in the paper).
>
> The profiling results align robustly with our theoretical analysis in Appendix A.4, confirming that PDT is scalable to large modern architectures. PDT introduces a linear space complexity $\mathcal{O}(N \times h)$, the relative memory overhead ratio decreases for larger tasks. This overhead is amortized in a large-scale training scenario where memory is dominated by activations and runtime is dominated by gradient computation. For ResNet-50 on ImageNet-1k, the relative PDT memory overhead ratio is only 13.0% (batch size 256). For ViT-Base on ImageNet, the relative PDT memory overhead ratio is 26.0%. All profiling experiments were conducted on a single GPU (Nvidia RTX A6000 or H100) to ensure accurate measurement of peak memory usage and timing without interference from distributed communication overhead. In practice, we used a batch size of 1800 (600 per GPU, trained on 3 GPUs) for ResNet-50 and ViT-base, so we estimate the overhead ratio should be even lower.
>
> As also noted in Sec. 5, reducing the memory footprint of Koopman approximations through streaming DMD is a promising direction for future work.
>
> **Table 8 Memory and Runtime overhead of PDT compared to baseline (AlexNet + CIFAR-10).**
> | Metric | Baseline | PDT |
> | :--- | :---: | :---: |
> | **GPU Memory Usage** | | |
> | - Peak GPU Memory (MB) | 735.9 | 9158.2 |
> | - Currently Allocated (MB) | 468.7 | 5078.0 |
> | - Reserved by Allocator (MB) | 994.0 | 11454.0 |
> | **PDT Memory Overhead** | | |
> | - Total PDT Overhead (MB) | — | 8422.4 |
> | - Overhead Ratio (%) | — | **1145%** |
> | **PDT Memory Overhead Breakdown** | | |
> | - Snapshot Storage (MB) | — | 1088.0 |
> | - SVD Workspace (MB) | — | 1740.9 |
> | - Other (MB) | — | 5593.4 |
> | **PDT Runtime Overhead (per epoch)** | | |
> | - Avg Time per Epoch (s) | 7.465 | 7.746 |
> | - Total PDT Overhead (s) | — | 0.333 |
> | - PDT Overhead Ratio (%) | — | **4.30%** |
> | **PDT Runtime Overhead Breakdown** | | |
> | - **Core PDT Operations (ms)** | — | 194.0 |
> | -- SVD decomposition (ms) | — | 98.3 |
> | -- Prediction (ms) | — | 88.1 |
> | -- Masking (ms) | — | 7.6 |
> | - Auxiliary Operations (ms) | — | 138.8 |
>
> **Table 9 Memory and Runtime overhead of PDT compared to baseline across different batch sizes (ResNet-50 + ImageNet-1k).**
> | Metric | Baseline (bs=64) | Baseline (bs=128) | Baseline (bs=256) | PDT (bs=64) | PDT (bs=128) | PDT (bs=256) |
> | :--- | :---: | :---: | :---: | :---: | :---: | :---: |
> | **GPU Memory Usage** | | | | | | |
> | - Peak GPU Memory (MB) | 3,062 | 5,792 | 11,258 | 4,528 | 7,255 | 12,721 |
> | - Currently Allocated (MB) | 400 | 436 | 512 | 2,439 | 2,436 | 2,439 |
> | - Reserved by Allocator (MB) | 3,404 | 6,766 | 13,424 | 7,392 | 10,864 | 15,298 |
> | **PDT Memory Overhead** | | | | | | |
> | - Total PDT Overhead (MB) | — | — | — | 1,466 | 1,463 | 1,463 |
> | - Overhead Ratio (%) | — | — | — | **47.9%** | **25.3%** | **13.0%** |
> | **PDT Memory Overhead Breakdown** | | | | | | |
> | - Snapshot Storage (MB) | — | — | — | 585 | 585 | 585 |
> | - SVD Workspace (MB) | — | — | — | 780 | 780 | 780 |
> | - Other (MB) | — | — | — | 101 | 98 | 98 |
> | **PDT Runtime Overhead (per epoch)** | | | | | | |
> | - Avg  Time per Epoch (s) | 813.9 | 653.1 | 626.5 | 804.0 | 651.5 | 635.3 |
> | - Total PDT Runtime Overhead (ms) | — | — | — | 90 | 90 | 110 |
> | - Overhead Ratio (%) | — | — | — | **0.011%** | **0.013%** | **0.017%** |
> | **PDT Runtime Overhead Breakdown** | | | | | | |
> | - **Core PDT Operations (ms)** | — | — | — | 51.5 | 50.7 | 51.1 |
> | -- SVD decomposition (ms) | — | — | — | 31.0 | 30.4 | 30.7 |
> | -- Prediction (ms) | — | — | — | 19.7 | 19.5 | 19.6 |
> | -- Masking (ms) | — | — | — | 0.8 | 0.8 | 0.8 |
> | - Auxiliary Operations (ms) | — | — | — | 38.5 | 39.3 | 58.9 |

---

> ### Author Response · Authors · 2025-11-24
> **Response to Reviewer SwkT (part3)**
>
> **Table 10: Memory and Runtime overhead of PDT compared to baseline (ViT-Base + ImageNet-1k).**
> | Metric | Baseline | PDT |
> | :--- | :---: | :---: |
> | **GPU Memory Usage** | | |
> | - Peak GPU Memory (MB) | 19,022 | 23,976 |
> | - Currently Allocated (MB) | 1,536 | 8,406 |
> | - Reserved by Allocator (MB) | 19,462 | 35,328 |
> | **PDT Memory Overhead** | | |
> | - Total PDT Overhead (MB) | — | 4,954 |
> | - Overhead Ratio (%) | — | **26.0%** |
> | **PDT Memory Overhead Breakdown** | | |
> | - Snapshot Storage (MB) | — | 1,981 |
> | - SVD Workspace (MB) | — | 2,642 |
> | - Other (MB) | — | 330 |
> | **PDT Runtime Overhead (per epoch)** | | |
> | - Avg Time per Epoch (s) | 831.3 | 832.1 |
> | - Total PDT Runtime Overhead (ms) | — | 877 |
> | - Overhead Ratio (%) | — | **0.11%** |
> | **PDT Runtime Overhead Breakdown** | | |
> | - **Core PDT Operations (ms)** | — | 193.2 |
> | -- SVD decomposition (ms) | — | 127.4 |
> | -- Prediction (ms) | — | 62.8 |
> | -- Masking (ms) | — | 3.0 |
> | - Auxiliary Operations (ms) | — | 683.6 |
>
> **Question 2: The relative importance of the two masking criteria**
>
> We thank the reviewer for this insightful suggestion. We have conducted a comprehensive ablation study for masking criterion, and added a detailed analysis in Appendix A.11 (Table 11 and Figs. 15-16).
>
> **Table 11: Comprehensive masking ablation results on AlexNet trained on CIFAR-10. Final accuracy reported as mean $\pm$ std across five seeds (successful runs only).**
> | Configuration | tau | LR=0.01 Success | LR=0.01 Acc (%) | LR=0.05 Success | LR=0.05 Acc (%) | Overall Success |
> | :--- | :---: | :---: | :---: | :---: | :---: | :---: |
> | Baseline (SGD) | — | 5/5 | 64.40 ± 0.71 | 5/5 | 78.68 ± 0.21 | **10/10** |
> | Accel Only | 3 | 5/5 | 58.04 ± 1.29 | 5/5 | 78.23 ± 0.64 | 16/20 |
> | Accel Only | 5 | 4/5 | 48.20 ± 8.19 | 2/5 | 10.00 ± 0.00 | |
> | Consistency Only | 3 | 0/5 | *Crashed* | 1/5 | 10.00 | 1/20 |
> | Consistency Only | 5 | 0/5 | *Crashed* | 0/5 | *Crashed* | |
> | Full PDT | 3 | 5/5 | **69.39 ± 0.79** | 5/5 | 78.98 ± 0.18 | **20/20** |
> | Full PDT | 5 | 5/5 | 69.10 ± 1.33 | 5/5 | **79.48 ± 0.20** | |
>
> **Relative Importance & Necessity of Both Criteria:** Our experiments on AlexNet (CIFAR-10) clearly decouple the roles of the two criteria:
> - Consistency Only (Eq. 7): This variant failed catastrophically, crashing in 19 out of 20 runs. This demonstrates that directional alignment alone is insufficient to constrain DMD predictions, which can be directionally correct but excessively large in magnitude.
> - Acceleration Only (Eq. 6): While it achieved some acceleration, it suffered from severe instability. For example, with LR=0.05 and $\tau=5$, the success rate dropped to 40%, and validation accuracy collapsed to random chance (10%). Without the directional check, "large but wrong" updates accumulate and destabilize training.
> - Full PDT (Both criteria): By combining both criteria, PDT achieved 100% robustness (20/20 success) and the highest accuracy, confirming that the two criteria are complementary and essential.
>
> **Masking Ratio Evolution:** You are absolutely correct that the masking ratio reflects the changing contributions of the criteria. Our new analysis (Fig. 16 in Appendix A.11) reveals an interesting dynamic: The Dynamic Consistency Criterion (Eq. 7) provides stable, stage-agnostic directional filtering and consistently rejects $\sim$50% of predictions (directionally mismatched) throughout training. The Acceleration Effectiveness Criterion (Eq. 6) shows a clear phase transition. In the early stage, the Lower Bound is more active (ensuring speedup). Crucially, in the late stage, the Upper Bound becomes the dominant rejection factor ($\sim$50% rejection rate). This confirms that as gradients shrink and oscillate near convergence, the upper bound prevents over-aggressive updates that could cause divergence and ensures stability.
>
> We have incorporated these findings into the revised paper to strengthen the theoretical justification of our masking strategy. Thank you again for these excellent questions!

---

> ### Author Response · Authors · 2025-11-24
> **Response to Reviewer SwkT (part4)**
>
> **Question 3: A rule of thumb for selecting additional parameters.**
>
> Please see the response to Weakness 2 above.
>
> **Question 4: Mismatch between Eq.6 and Eq.8.**
>
> Please see the response to Weakness 1 above.
>
> **Question 5: How many batches were in an epoch?**
>
> Thank you for this question. The number of skipped gradient steps is determined by the dataset size and the batch size. To provide a clear picture of the acceleration magnitude, we have summarized the Steps per Epoch and the Total Skipped Gradient Steps (per single PDT prediction with our default prediction steps $\tau=5$) for our key experiments (in Table 1) in the table below.
>
> **Dataset Statistics:**
> - CIFAR-10: 50,000 training images
> - ImageNet-1k: 1,281,167 training images
>
> Steps per Epoch: $\lceil \text{Dataset Size} / \text{Total Batch Size} \rceil$
>
> Skipped Steps per Prediction: $\text{Steps per Epoch} \times \tau$
>
> **Table: Gradient Steps and Skipped Steps per Prediction Block ($\tau=5$)**
> | Dataset | Model | Global Batch Size | Steps per Epoch | Skipped Steps (per PDT prediction) |
> | :--- | :--- | :---: | :---: | :---: |
> | CIFAR-10 | AlexNet / FCN | 256 | 196 | **980** |
> | ImageNet-1k | ResNet-50 / ViT-Base | 1,800 (600 x 3 GPUs) | 712 | **3,560** |
>
> As shown in the above table, PDT bypasses a significant number of iterative updates (e.g., ~3,560 SGD steps on ImageNet-1k) in a single prediction step, which explains the substantial wall-clock time reduction observed in our results. Note that due to the acceleration effectiveness criterion (Eq.6)  and the dynamic consistency criterion (Eq.7) of PDT, it only performs predictive updates on the weight that meet the masking criteria.

---

### Official Review · Reviewer_efcL · 2025-10-31

**Soundness:** 2
**Presentation:** 1
**Contribution:** 2
**Rating:** 2
**Confidence:** 3

**Summary:**

This paper leverages the predictive capability of KOT and introduces a framework PDT to accelerate neural network training by allowing weight updates to bypass standard gradient steps and use predicted weights instead. A dynamic consistency analysis is proposed to ensure stability when scaling to larger networks, and experiments are conducted to demonstrate empirical gains.

**Strengths:**

1. The paper explores an interesting intersection between optimal control and DNNs training, highlighting a potentially useful connection between the two fields.
2. The authors conduct experiments across multiple optimizers, architectures, and datasets to demonstrate the generality and potential improvements of the proposed method.

**Weaknesses:**

- The paper’s writing and logical flow make it difficult to understand what the proposed method actually does. The main text introduces several abstract notations (\(x_i, w_i, W_i, g, K, T\)) with minimal explanation or connection to gradient-based optimization. For example, the authors use \(x\), \(w\), and \(W\) all as inputs to \(g\), which easily confuses readers. The overall presentation would greatly benefit from clearer definitions, consistent notation, and explicit mapping to standard training dynamics.

- Most essential implementation details about how the Koopman operator is estimated, how prediction integrates with standard optimizers, and how the dynamic consistency mask is computed—are deferred to the appendix. Since reading appendices is optional during review, this omission makes it impossible to fully evaluate the method’s soundness, novelty, and overall contribution from the main paper alone.

- The toy example provided does not effectively illustrate why or how the method applies to general neural network training. Its formulation lacks connection even to a simple MLP and therefore fails to provide intuition or justification for why Koopman-based prediction should improve training efficiency.

- The paper mentions that predicted updates are applied "once a while," but it does not specifies or studies, either theoretically or empirically, how often these predictions should be used. This frequency is critical for both computational cost and model performance. Without an ablation study on this factor, it is difficult to assess the reliability of the reported improvements.

- The "mask," which seems to determine when to apply predicted updates, is central to the method’s stability but is not properly introduced or explained in the main text. It is defined only in the appendix as a binary gate based on heuristics. The paper does not justify why a mask is necessary, why it must be binary, or how this design choice impacts performance or stability.

- The method requires repeated SVD to estimate the Koopman operator, yet the paper provides no analysis of the associated computational overhead. There is no information about matrix sizes, decomposition frequency, or runtime breakdown. Since SVD can have cubic complexity, this omission raises serious concerns about scalability—especially given the claim of “10–40% faster convergence.”

**Questions:**

- Is $\mathcal{K}=\sum_k \lambda_k \phi_k $? Is $c_k = \langle \phi_k, g\rangle$?
- What is data matrices $W_i$ and $W_{i+1}$ in neural networks? Are they neural network weights parameters?
- When you say $A$ can be solved but you actually mean approximate? cuz equation (4) is not equal
- how frequent you apply the prediciton step? does that depends on the optimziers, network achitecures, datasets?

### **Questions for the Authors**

- Is $\mathcal{K}=\sum_k \lambda_k \phi_k $? Is $c_k = \langle \phi_k, g\rangle$?

- What are the data matrices $W_i$ and $W_{i+1}$ in the context of neural networks? Are they weight parameters?

- How frequently is the Koopman-based prediction step applied during training? Is the prediction frequency fixed or adaptive? Does it depend on the choice of optimizer, network architecture, or dataset?

- How is the **binary mask** implemented in practice? Is it computed at every prediction?

- What is the computational cost of performing the SVD at each prediction step? How large are the snapshot matrices used in your experiments?

- How sensitive is PDT to the number of snapshots used for Koopman estimation? Have you tested different window lengths or ranks for the decomposition?

---

> ### Author Response · Authors · 2025-11-23
> **Response to Reviewer efcL (part1)**
>
> Thanks for your valuable comments. We will address your concerns one by one in the following.
>
> **Weakness 1: Regarding the writing, logical flow, and notations.**
>
> The entire Sec. 1 is to establish the connection between the proposed accelerated training and gradient-based optimizations. The first paragraph in Sec. 1 explains the “fundamental limitations” of gradient-based optimization. The second paragraph describes “differential learning” and related works where different parts of the network can be trained differently. The third paragraph introduces “predictive training” and related works, where weights are predicted instead of being learned through gradient descent. The sentence in italic “the DNN training process itself is a nonlinear dynamical system acting upon the high-dimensional weight space”. The fourth paragraph discusses bottlenecks of existing predictive training approaches that apply Koopman Operator Theory (KOT) to predict ALL weights. The rest of Sec. 1 discusses the “key observation” and proposes “predictive differential training” (PDT) where prediction is incorporated but only to selective weights determined by a mask.
>
> The notations, $x_i, w_i, W_i, g, K$, and $T$ are all explained in Sec. 2 (Background) and Sec. 3 (Method). Since we adopt the Koopman Operator Theory to predict the next state of a nonlinear dynamical system, the background section mainly focuses on explaining KOT. $T$ is the mapping (or transfer) function that maps from the current state of the system ($x_i$) to the next (or future) state ($x_{i+1}$); hence $x_{i+1} = T(x_i)$.  $g(.)$ is the observable (or measurement) function of the system state. As an analogy, $g(.)$ is like taking the biometrics of the human body, including, for example, heart beat, blood pressure–measurements of the human body (which is a very complex dynamical system). Since it is difficult to learn the actual state of a nonlinear dynamical system, KOT uses measurements to “indirectly” learn the state. $K$ is an approximation to $T$; hence $g(x_{i+1}) = Kg(x_i)$ as compared to $x_{i+1} = T(x_i)$ early on.
>
> Starting from the DMD paragraph, we discuss specifically under the context of DNN training. As we explained above (and in Sec. 1), we are treating the DNN training process as a dynamical system whose input is the current weights, $w_i$ and the output is the next (predicted) weights, $w_{i+1}$. Hence we can regard the “weights” as the observables (or measurements) of the DNN network. Since the network is fully represented by its weights, we consider this system fully observable, that is, $w_i = g(w_i)$. $W_i$ is a matrix with each column representing the weights vector from previous few epochs. $W_i$ needs to be constructed so that we can adopt the dynamic mode decomposition (DMD) to find the mapping function.
>
> **Weakness 2: Regarding the implementation details of PDT.**
>
> The main paper does contain the complete workflow and visual explanation (Sec. 3.1, Figure 3) of the proposed PDT approach. The full mathematical formulation of the masking strategy from dynamic consistency analysis (Equations 6-7, detailed in Section 3.2) and the DMD prediction equation (Equation 5) are also provided, showing how spectral decomposition enables multi-step forecasting. Comprehensive experimental setup and results are presented in Section 4, Tables 1-3. We only defer to the appendix some additional results, and in-depth analyses of efficiency and convergence, including, for example, A.1 Convergence path of the toy example; A.2 Pseudocode of the PDT algorithm; A.3 Computational efficiency analysis; A.4 Computational complexity analysis; A.5 Complete proof of convergence; A.6 More results on various network structures; A.7 Mask distribution analysis; A.8 Effect of non-i.i.d. training data; A.9 Hyperparameter analysis; A.10 Profiling and overhead analysis.
>
> **Weakness 3: Regarding the purpose of the toy example**
>
> The toy example in Section 3 was designed to provide intuition for the principle of differential learning—that strategically accelerating a subset of variables can speed up the learning process. We deliberately chose a simple 6-variable “root-finding problem” using gradient descent. We show that the usage of a larger learning rate on the 3 out of 6 variables can speed up problem solving.

---

> ### Author Response · Authors · 2025-11-23
> **Response to Reviewer efcL (part2)**
>
> **Weakness 4: Regarding the prediction frequency.**
>
> We respectfully note that the prediction interval $T_i$ is a crucial parameter and was specified multiple times in the paper. In Section 4.1, we state, “In all experiments, we use the past five epochs to form the snapshot **with a one-epoch interval to predict weights** in the next five steps. **Prediction starts** from the 5th epoch.” Each and every experimental result is a comparison between “baseline” (i.e., without prediction including SGD, Momentum, AdamW, RMSprop, Shampoo, and LAMB) and “PDT” (i.e., with prediction embedded in the baselines). Please see Tables 1-3 in the main paper.
>
> In addition, in Appendix A.9 and Figure 14(b), we provided an ablation study about PDT-related hyperparameters, including the prediction interval $T_i$. The results show that overly frequent or overly sparse predictions will affect performance. Our default setting is a trade-off between efficiency and stability. We stated that “The PDT-related hyperparameters mentioned were set to prediction step=5, prediction interval=1, start epoch=5, and past snapshot counts=5.” These values are robust across optimizers, architectures, and datasets.
>
> **Weakness 5: Regarding the design of the masking strategy.**
>
> In Section 3.2, we provided a detailed description of how our masking strategy was designed. The two criteria (Eqs. 6-7) provide orthogonal benefits: acceleration effectiveness ensures speedup is meaningful (without this, we might accept predictions that barely move weights, providing no acceleration), while dynamic consistency ensures directional alignment (without this, we might accept predictions moving in the wrong direction, causing divergence). Both are necessary conditions for safe acceleration.
>
> The discrete nature of our update decision fundamentally justifies the binary mask design. For each weight parameter, we must choose between accepting the DMD prediction (potentially faster convergence) or falling back to the SGD update (guaranteed stability). There is no meaningful semantic interpretation for "partial acceptance" in this context—we cannot use 0.7 of a predicted weight and 0.3 of an SGD weight because this would create directional ambiguity about which optimization trajectory we are following. The binary decision ensures that each parameter follows a consistent update direction, which is critical for convergence guarantees. Our convergence analysis in Appendix A.5 relies on each update being either a properly bounded DMD prediction satisfying Eqs. 6-7, or a standard gradient step with known convergence properties. Mixing predictions and gradients for the same parameter would introduce uncertainty about whether the combined update satisfies descent conditions.
>
> **Weakness 6: Regarding the computational overhead introduced by SVD.**
>
> We provided detailed computational overhead analysis in Appendix A.4. Appendix A.4 analyzes computational complexity, explicitly stating "The primary computational burden arises from performing SVD on this matrix with a complexity of $O(N \times h^2)$" and explaining why this is manageable: $N$ significantly exceeds $h$ ($h$ typically equals 5 while $N$ reaches millions or billions), and SVD occurs infrequently (once per epoch versus hundreds of batches per epoch).
>
> We have also added a detailed "Profiling and Overhead Analysis" section in Appendix A.10. We conduct comprehensive profiling experiments across three diverse architectures: AlexNet (CNN, 57M parameters) on CIFAR-10 (Table 8), ResNet-50 (deep CNN, 25.6M parameters) on ImageNet-1k (Table 9), and ViT-Base (Transformer, 86.4M parameters) on ImageNet-1k (Table 10). We measure wall-clock time, GPU memory usage, and FLOPs for both the baseline optimizer and PDT. Please refer to Tables 8-10 in the revised paper, or you can find them in our comments for Reviewer sCUV.
>
> **Question 1.**
>
> Regarding $\mathcal{K}$: In Eq. 2, we follow the standard Koopman mode decomposition. The operator $\mathcal{K}$ acts on the observable $g(x)$ (which is the identity function $g(w)=w$ in our case). The spectral expansion describes the evolution of this observable: $\mathcal{K} g(x_i) = \sum_{k=1}^{\infty} \lambda_k \phi_k(x_i) c_k$, where $\lambda_k$ are the eigenvalues and $\phi_k$ are the eigenfunctions.
>
> Regarding $c_k$: $c_k$ represents the Koopman modes (amplitudes). The reviewer asks if $c_k = \langle \phi_k, g \rangle$. This equality would hold only if the eigenfunctions $\{\phi_k\}$ formed an orthonormal basis. However, in the context of Dynamic Mode Decomposition (DMD), the derived modes are generally not orthogonal.
> Therefore, we compute $c_k$ by projecting the initial state onto the modes using the pseudo-inverse, not the inner product. This is explicitly defined in our Eq. 5.

---

> > ### Comment · Reviewer_efcL · 2025-11-26
> >
> > The reviewer thanks the authors for their thoughtful rebuttal. I have read the response and **reread** the manuscript carefully. While I find the overall idea interesting and potentially useful, I still cannot fully understand several essential implementation details. I hope the authors can clarify the following points, because they seem fundamental to understanding the proposed method.
> >
> > 1. The role of $A=W_{i+1}W_i^{\dagger}$ is unclear. This object is introduced in the main text but never used again in the paper and algorithm. Equation (5) and Algorithm 1 instead uses $\Phi$ and $\Lambda$, without explaining how these matrices are computed from $W_{i+1}$ and $W_i$.
> >
> > 2. The authors state that an eigendecomposition of $A$ is performed, but $A$ is not symmetric or normal generally. Thus, in general, we cannot write $A=\Phi \Lambda \Phi'$. Thus, it is unclear: how exactly are $\Phi$ and $\Lambda$ obtained? How do $\Phi$ and $\Lambda$ relate to the SVD factors $U$, $\Sigma$, $V$ of $W_i$? How is $\Phi^{\dagger}$ computed in practice for use in equation (5)?
> >
> > 3. Efficiency claims are not justified. The complexity analysis focuses only on the SVD of $W_i$, but does not describe how the SVD leads to $\Phi$ and $\Lambda$, or why the overall computation of Eq. (5) is efficient in practice. Without a clear derivation of $\Phi$ and $\Lambda$, the practical computational gains of the method are not fully explained.
> >
> > All of this seems to rely on DMD, but DMD is not clearly introduced in neither the main text nor explained in Appendix. As a result, the logic flow between
> > $$
> > (W_i, W_{i+1}) \to A\to (\Phi, \Lambda)
> > $$
> > is not made explicit anywhere in the paper.
> >
> > I believe the method is interesting and potentially valuable to a broad ML audience, and I am willing to raise my score. However, in its current form, several key implementation details are not described clearly enough for general ML readers. I kindly ask the authors to clarify these steps in the main text and/or Appendix, so that the paper is understandable to a general ML audience rather than only to experts familiar with DMD techniques from fluids or control theory.

---

> > > ### Author Response · Authors · 2025-11-27
> > > **Response to Reviewer efcL**
> > >
> > > We sincerely thank the reviewer for the thorough re-evaluation and for identifying the missing links in our method description. Due to page limit, in the original text, we only get to refer the readers to [Kutz et al. (2016)] for details of DMD. But we do agree that we should provide the connection from $(W_i, W_{i+1}) \to A \to (\Phi, \Lambda)$ more explicitly.
> > >
> > > To address this, we have added a new section **Appendix A 2.1: Derivation of the DMD Algorithm**, which provides a very detailed, step-by-step derivation connecting the snapshots to the prediction equation. We clarify your specific questions below:
> > >
> > > > 1. The Role of $A$ and the Derivation of $\Phi, \Lambda$
> > >
> > > We clarify that we never directly compute the high-dimensional matrix $A = W_{i+1} W_i^{\dagger}$ (which would be $N \times N$). Instead, we employ the *Standard DMD* to compute the spectral decomposition in a low-rank subspace.
> > >
> > > - **Step 1 (SVD)**: We first compute the reduced SVD of the snapshot matrix $W_i \approx U \Sigma V^T$, where $U \in \mathbb{R}^{N \times r}$ are the proper orthogonal decomposition (POD) modes. Here $r \le h$ is the truncation rank.
> > > - **Step 2 (Projection)**: Instead of computing the high-dimensional operator $A = W_{i+1}W_i^\dagger = W_{i+1} V \Sigma^{-1} U^T$, we compute a small proxy matrix $\tilde{A} = U^T A U = U^T (W_{i+1} W_i^{\dagger}) U = U^T W_{i+1} V \Sigma^{-1}$ of size $r \times r$ (where $r \le h$). This $\tilde{A}$ represents the operator $A$ projected onto the subspace spanned by $U$.
> > > - **Step 3 (Eigendecomposition)**: We compute the eigendecomposition of this small matrix: $\tilde{A} \Psi = \Psi \Lambda$, where $\Lambda = \text{diag}(\lambda_1, \dots, \lambda_r)$ contains the eigenvalues used in Eq. (5) (which approximate the eigenvalues of the full operator $A$), and $\Psi \in \mathbb{C}^{r \times r}$ contains the eigenvectors of $\tilde{A}$.
> > > - **Step 4 (Modes $\Phi$)**: To reconstruct the high-dimensional modes, we use the *Standard DMD* formulation: $\Phi = U \Psi$. This explicitly defines how $\Phi$ is obtained from the SVD factors and data matrices.
> > >
> > > > 2. Handling Non-Symmetry of $A$
> > >
> > > The review is correct that $A$ is generally not symmetric, meaning the modes $\Phi$ are not orthogonal, so $\Phi^{\dagger} \neq \Phi^T$.
> > >
> > > In our implementation (and Eq. 5), the term $\mathbf{c} = \Phi^{\dagger} \mathbf{w}\_i$ represents the projection of the current weights onto the DMD modes (i.e., the vector of mode amplitudes). Since $\Phi$ is a "tall and skinny" matrix ($N \times r$), we compute this by solving the overdetermined linear least-squares problem:
> > > $$\min_{\tilde{\mathbf{c}}} \| \mathbf{w}_i - \Phi \tilde{\mathbf{c}} \|_2$$
> > >
> > > This provides the optimal projection of the current weights onto the non-orthogonal DMD modes. The final prediction is then computed as $w^{pred}_{i+\tau} = \text{Re}(\Phi \Lambda^\tau \mathbf{c})$.
> > >
> > > > 3. Justification of Efficiency Claims
> > >
> > > The complexity analysis in the original paper was indeed too brief. In the revised Appendix A.4, we provide a detailed breakdown. The computational cost is dominated by matrix multiplications involving the parameter dimension $N$ and the snapshot count $h$ (e.g., calculating $\tilde{A}$ and $\Phi$).
> > >
> > > - SVD of $W_i$: $\mathcal{O}(N \times h^2)$
> > > - Computing $\tilde{A}$: $\mathcal{O}(N \times h^2)$
> > > - Eigendecomposition of $\tilde{A}$: $\mathcal{O}(h^3)$
> > > - Computing Modes $\Phi$: $\mathcal{O}(N \times h^2)$
> > > - Prediction (Solve $\Phi^{\dagger}$): $\mathcal{O}(N \times h^2)$
> > >
> > > The total complexity sums to $\mathcal{O}(N \times h^2)$. Since we fix the past snapshot counts to a small integer ($h=5$) while $N$ can be millions or even billions, the quadratic impact of $h$ remains manageable relative to $N$. Thus, the algorithm scales linearly with $N$. This mathematically justifies why the wall-clock overhead is minimal in practice. We have also added a detailed "Profiling and Overhead Analysis" section in Appendix A.10.  For larger datasets like ImageNet-1k, where time per epoch is longer, the runtime overhead becomes negligible. Our profiling shows a PDT runtime overhead ratio of only 0.017% for ResNet-50 (in Table 9, batch size 256) and 0.11% for ViT-base (in Table 10).
> > >
> > > We have revised Section 3.2 in the main text (see highlighted) to better connect $A$ with $\Phi$ and $\Lambda$.  We have also explicitly referenced Appendix A 2.1 for the mathematical derivation of DMD, ensuring the logic flow is clear to general ML readers. We hope this detailed derivation connects the missing link and clarifies the practical implementation of our method.
> > >
> > > Please let us know if there are any other descriptions in the text that are unclear. Thanks again for the thorough review that definitely helps improve the readability of the paper.

---

> > > > ### Comment · Reviewer_efcL · 2025-11-27
> > > >
> > > > The reviewer thanks the authors for the additional clarification. I am satisfied with the rebuttal and have raised my score to recommend acceptance.

---

> > > > > ### Author Response · Authors · 2025-11-28
> > > > >
> > > > > We sincerely thank the reviewer for the positive evaluation and for recommending acceptance. We are glad that our clarifications addressed your concerns. Your constructive feedback has significantly improved the clarity of our manuscript.

---

> ### Author Response · Authors · 2025-11-23
> **Response to Reviewer efcL (part3)**
>
> **Question 2: Regarding data matrices $W\_i$ and $W\_{i+1}$**
>
> Yes, they are weight parameter matrices. As defined in Eq. 4, $W_i$ is the snapshot matrix constructed by stacking flattened neural network weight vectors from $h$ consecutive epochs: $W\_i = [\mathbf{w}\_t, \mathbf{w}\_{t+1}, \dots, \mathbf{w}\_{t+h-1}]$, where each column $\mathbf{w}$ represents the entire set of model parameters at a specific epoch. Similarly, $W\_{i+1}$ is the time-shifted version: $W\_{i+1} = [\mathbf{w}\_{t+1}, \mathbf{w}\_{t+2}, \dots, \mathbf{w}\_{t+h}]$. These matrices are the input for the DMD algorithm to compute the linear operator that best approximates the weight evolution.
>
> **Question 3: Regarding "solved" vs. "approximate" in Eq. 4**
>
> The Koopman operator is infinite-dimensional, and DMD provides a finite-dimensional linear approximation of these non-linear dynamics. When we state "A can be solved," we refer to the optimization problem. Specifically, we solve for the matrix $A$ that minimizes the Frobenius norm of the approximation error: $\min\_A \| W\_{i+1} - A W\_i \|\_F$.The solution to this least-squares problem is exact and is given by $A = W\_{i+1} W\_i^\dagger$ (using the pseudo-inverse), even though the relationship $W\_{i+1} \approx A W\_i$ remains an approximation of the underlying system.
>
> **Question 4: Regarding the prediction frequency.**
>
> As detailed in Appendix A.9, the prediction frequency is controlled by the parameter $T_i$ (prediction interval). Our default and robust setting is $T_i=1$. We found that a fixed interval of $T_i=1$ (combined with our masking strategy) yields robust performance across different optimizers (e.g., SGD, Adam, RMSprop), architectures (CNNs, ViTs), and datasets (CIFAR-10, ImageNet-1k). We analyze the sensitivity of this hyperparameter in Appendix A.9 and Figure 14(b). We also have proposed "Rule of Thumb" guidelines to help find the appropriate hyperparameters for different scenarios in Appendix A.9. While the PDT framework allows adaptive intervals, we can enable this function when the validation loss is too high or when we want to reduce additional costs.
>
> **Question 5: Regarding the binary mask implementation.**
>
> The mask is indeed computed at every prediction step. Before applying any predicted weights, we calculate the mask $M$ element-wise. For a prediction step $\tau$, we compute the mask $m_j \in \{0, 1\}$ for each parameter $j$ by checking Acceleration Effectiveness (Eq. 6) and Dynamic Consistency (Eq. 7), where $j$ is the index for each element in the weight vector. Only if both conditions are met is $m_j$ set to 1.
>
> **Question 6: Regarding SVD computational cost and matrix sizes.**
>
> As detailed in Appendix A.4, the snapshot matrices have dimensions $N \times h$, where $N$ is the number of model parameters and $h$ is the number of snapshots. For example, for ResNet-50, $N \approx 25.5 \text{ million}$.In our experiments, $h$ is kept small (typically $h=5$). The complexity of the randomized SVD used in DMD is dominated by $\mathcal{O}(N \times h^2)$. Since $h \ll N$ (5 vs. 25.5 million), the cost is linear with respect to the number of network parameters, similar to a standard gradient update. As shown in Appendix A.3 and Figure 9, the wall-clock time overhead of this SVD calculation is negligible compared to the time saved by skipping gradient iterations. We have also added a detailed "Profiling and Overhead Analysis" section in Appendix A.10.
>
> **Question 7: Regarding sensitivity to snapshots ($h$) and Rank.**
>
> We conducted a sensitivity analysis on the number of snapshots $h$ in Appendix A.9 and Figure 14(d). We observed a "sweet spot" around $h=5$. Suppose $h$ is too small ($h = 3$), the snapshot matrix contains insufficient information to capture the dynamics. If $h$ is too large ($h = 10$), the system includes too many "stale" weights from much earlier in training and introduces additional overhead. Regarding the Rank, since $h$ is very small ($h=5$), the rank of the approximation is naturally bounded by $h$. We effectively use a full-rank decomposition of the snapshot matrix, but this is a low-rank approximation ($r \le 5$) relative to the full parameter space $N$. Truncating the rank further (e.g., rank < $h$) is unnecessary given that $h$ is already a compact window.

---

### Official Review · Reviewer_sCUV · 2025-11-02

**Soundness:** 3
**Presentation:** 3
**Contribution:** 3
**Rating:** 4
**Confidence:** 4

**Summary:**

The paper proposed PDT that selectively applies updates to accelerate the training. Their method can be integrated with diverse optimizers. The authors provides diverse experiments on synthetic data and realistic dataset that their methods can accelerate the training progess without sacrificing model performance.

**Strengths:**

1. Their method can be integrated with lots of classic optimizers as a plug-in tool.
2. The seletive design ensures the stability of training
3. The predictive power of PDT accelerates the convergence (in iteration-wise sense).

**Weaknesses:**

1. The measurement of runtime is not well explained. I'm not sure if the total runtime includes the prediction time for accelerated training for PDT.
2. The training results cannot match SOTA sometimes. For example, in figure 10d, the training on ImageNet-1K using ViT-Base can achieve ~76% validation accuracy but in the figure the best is below 70%. Similar for the training of ImageNet-1K on ResNet50.
3. The complexity analysis didn't cover the space analysis. So I'm not sure if it can extends to large-scale trainings as in these trainings space is also very critical.

**Questions:**

1. In figure 4, is the base optimizer SGD for PDT?
2. In line 1041, the citation is missing: "The PDT-related hyperparameters mentioned in Sec. ?? ".

---

> ### Author Response · Authors · 2025-11-23
> **Response to Reviewer sCUV (Part1)**
>
> Thanks for your valuable review. We will address your concerns one by one in the following.
>
> **Weakness 1: The measurement of runtime.**
>
> Sorry for the confusion. The short answer is yes – the runtime does include the prediction time. All runtime measurements reported in our paper (Tables 1-7) are total wall-clock times, including the complete computational overhead of PDT. This encompasses SVD computation, mask computation, weight prediction, all associated data structure operations and memory transfers, and the standard baseline optimization steps (e.g., SGD forward/backward passes). We took a straightforward approach to measure the runtime: we recorded wall-clock time from the start of training (epoch 0) to the end (last epoch). The metric "Time to Baseline Best Train Loss" (TTB-Loss) captures the actual elapsed time, including all PDT overhead. For example, when we report that AlexNet on CIFAR-10 achieves a 37% runtime reduction on training loss (in Table 1), this means PDT reaches the baseline's lowest loss value in 37% less wall-clock time despite performing all additional prediction operations. In the revised paper, we have clarified the definition of "Runtime" in Section 4.1. Please see the highlighted.
>
> To further address the concern raised, **we have added a detailed "Profiling and Overhead Analysis" section in Appendix A.10**. Our profiling on AlexNet with CIFAR10 (Table 8) shows that the total PDT overhead is approximately 333ms per prediction epoch (only 4.30% of the epoch time). For larger datasets like ImageNet-1k where time per epoch is longer, the runtime overhead becomes negligible. Our profiling shows a PDT runtime overhead ratio of only 0.017% for ResNet-50 (in Table 9) and 0.11% for ViT-base (in Table 10). The overall convergence speedup easily outweighs this slight per-epoch overhead.
>
> Please refer to Appendix A.10.1 in the paper to view the explanations for each term in the following tables.
>
> **Table 8 Memory and Runtime overhead of PDT compared to baseline (AlexNet + CIFAR-10).**
> | Metric | Baseline | PDT |
> | :--- | :---: | :---: |
> | **GPU Memory Usage** | | |
> | - Peak GPU Memory (MB) | 735.9 | 9158.2 |
> | - Currently Allocated (MB) | 468.7 | 5078.0 |
> | - Reserved by Allocator (MB) | 994.0 | 11454.0 |
> | **PDT Memory Overhead** | | |
> | - Total PDT Overhead (MB) | — | 8422.4 |
> | - Overhead Ratio (%) | — | **1145%** |
> | **PDT Memory Overhead Breakdown** | | |
> | - Snapshot Storage (MB) | — | 1088.0 |
> | - SVD Workspace (MB) | — | 1740.9 |
> | - Other (MB) | — | 5593.4 |
> | **PDT Runtime Overhead (per epoch)** | | |
> | - Avg Time per Epoch (s) | 7.465 | 7.746 |
> | - Total PDT Overhead (s) | — | 0.333 |
> | - PDT Overhead Ratio (%) | — | **4.30%** |
> | **PDT Runtime Overhead Breakdown** | | |
> | - **Core PDT Operations (ms)** | — | 194.0 |
> | -- SVD decomposition (ms) | — | 98.3 |
> | -- Prediction (ms) | — | 88.1 |
> | -- Masking (ms) | — | 7.6 |
> | - Auxiliary Operations (ms) | — | 138.8 |
>
> **Table 9 Memory and Runtime overhead of PDT compared to baseline across different batch sizes (ResNet-50 + ImageNet-1k).**
> | Metric | Baseline (bs=64) | Baseline (bs=128) | Baseline (bs=256) | PDT (bs=64) | PDT (bs=128) | PDT (bs=256) |
> | :--- | :---: | :---: | :---: | :---: | :---: | :---: |
> | **GPU Memory Usage** | | | | | | |
> | - Peak GPU Memory (MB) | 3,062 | 5,792 | 11,258 | 4,528 | 7,255 | 12,721 |
> | - Currently Allocated (MB) | 400 | 436 | 512 | 2,439 | 2,436 | 2,439 |
> | - Reserved by Allocator (MB) | 3,404 | 6,766 | 13,424 | 7,392 | 10,864 | 15,298 |
> | **PDT Memory Overhead** | | | | | | |
> | - Total PDT Overhead (MB) | — | — | — | 1,466 | 1,463 | 1,463 |
> | - Overhead Ratio (%) | — | — | — | **47.9%** | **25.3%** | **13.0%** |
> | **PDT Memory Overhead Breakdown** | | | | | | |
> | - Snapshot Storage (MB) | — | — | — | 585 | 585 | 585 |
> | - SVD Workspace (MB) | — | — | — | 780 | 780 | 780 |
> | - Other (MB) | — | — | — | 101 | 98 | 98 |
> | **PDT Runtime Overhead (per epoch)** | | | | | | |
> | - Avg  Time per Epoch (s) | 813.9 | 653.1 | 626.5 | 804.0 | 651.5 | 635.3 |
> | - Total PDT Runtime Overhead (ms) | — | — | — | 90 | 90 | 110 |
> | - Overhead Ratio (%) | — | — | — | **0.011%** | **0.013%** | **0.017%** |
> | **PDT Runtime Overhead Breakdown** | | | | | | |
> | - **Core PDT Operations (ms)** | — | — | — | 51.5 | 50.7 | 51.1 |
> | -- SVD decomposition (ms) | — | — | — | 31.0 | 30.4 | 30.7 |
> | -- Prediction (ms) | — | — | — | 19.7 | 19.5 | 19.6 |
> | -- Masking (ms) | — | — | — | 0.8 | 0.8 | 0.8 |
> | - Auxiliary Operations (ms) | — | — | — | 38.5 | 39.3 | 58.9 |

---

> ### Author Response · Authors · 2025-11-23
> **Response to Reviewer sCUV (Part2)**
>
> **Table 10: Memory and Runtime overhead of PDT compared to baseline (ViT-Base + ImageNet-1k).**
> | Metric | Baseline | PDT |
> | :--- | :---: | :---: |
> | **GPU Memory Usage** | | |
> | - Peak GPU Memory (MB) | 19,022 | 23,976 |
> | - Currently Allocated (MB) | 1,536 | 8,406 |
> | - Reserved by Allocator (MB) | 19,462 | 35,328 |
> | **PDT Memory Overhead** | | |
> | - Total PDT Overhead (MB) | — | 4,954 |
> | - Overhead Ratio (%) | — | **26.0%** |
> | **PDT Memory Overhead Breakdown** | | |
> | - Snapshot Storage (MB) | — | 1,981 |
> | - SVD Workspace (MB) | — | 2,642 |
> | - Other (MB) | — | 330 |
> | **PDT Runtime Overhead (per epoch)** | | |
> | - Avg Time per Epoch (s) | 831.3 | 832.1 |
> | - Total PDT Runtime Overhead (ms) | — | 877 |
> | - Overhead Ratio (%) | — | **0.11%** |
> | **PDT Runtime Overhead Breakdown** | | |
> | - **Core PDT Operations (ms)** | — | 193.2 |
> | -- SVD decomposition (ms) | — | 127.4 |
> | -- Prediction (ms) | — | 62.8 |
> | -- Masking (ms) | — | 3.0 |
> | - Auxiliary Operations (ms) | — | 683.6 |
>
> **Weakness 2: The training results cannot match SOTA sometimes.**
>
> We appreciate the reviewer’s observation. We want to clarify that our experiments (Fig. 10) are not intended to reproduce SOTA accuracy, but to **evaluate PDT relative to its baseline under exactly the same training configuration**. For ResNet-50 (Fig. 10c), our baseline achieves approximately 70% accuracy compared to the typical SOTA of approximately 76% top-1 accuracy. This gap exists because we used basic data augmentation without modern techniques like AutoAugment, RandAugment, CutMix, or Mixup, employed no label smoothing or additional regularization, and used standard SGD with momentum and basic cosine annealing without sophisticated warmup schedules or learning rate tuning. Similarly, for ViT-Base (Figure 10d), we used a batch size of 1800 ($600 \times 3$) instead of the SOTA's typical 1024-4096 with linear learning rate scaling, and no ViT-specific training enhancements such as LayerScale or stochastic depth.
>
> Our focus was on demonstrating relative improvement: showing that PDT accelerates convergence compared to baseline under identical conditions (for any given training configuration), maintains or improves final accuracy relative to baseline under those conditions, and works robustly across different architectures and scales. We will clarify this motivation and evaluation setup in the revision. If the reviewer considers SOTA performance critical, we are willing to conduct additional experiments using fully tuned modern training recipes to demonstrate that PDT is compatible with and beneficial under state-of-the-art training practices.
>
> **Weakness 3: The complexity analysis didn't cover the space analysis.**
>
> We thank the reviewer for raising this point. The memory/space complexity of PDT comes primarily from storing the past $h$ parameter snapshots used to form the weight trajectory matrix for the Koopman/DMD approximation, which scales as $\mathcal{O}(N \times h)$, where $N$ denotes the number of model parameters and $h$ denotes the number of past epochs retained. In all experiments, we intentionally keep $h$ small (set to 5), which limits the additional memory overhead. These snapshots are stored only temporarily and are overwritten after each DMD computation, so the space cost does not accumulate over training. The successful experiments on ViT-Huge (632M parameters) showcased the feasibility of the additional memory requirement from DMD.
>
> **We have added the above space complexity analysis in Appendix A.4 (see highlighted)**. **We have also added detailed memory profiling tables in Appendix A.10** showing peak GPU memory usage, PDT memory overhead, and detailed overhead for each PDT component. (Please see Tables 8-10 above and in the paper).
>
> The profiling results align robustly with our theoretical analysis in Appendix A.4, confirming that PDT is scalable to large modern architectures. PDT introduces a linear space complexity $\mathcal{O}(N \times h)$, the relative memory overhead ratio decreases for larger tasks. This overhead is amortized in a large-scale training scenario where memory is dominated by activations and runtime is dominated by gradient computation. For ResNet-50 on ImageNet-1k (Table 9), the relative PDT memory overhead ratio is only 13.0% (batch size 256). For ViT-Base on ImageNet-1k (Table 10), the relative PDT memory overhead ratio is 26.0%. All profiling experiments were conducted on a single GPU (Nvidia RTX A6000 or H100) to ensure accurate measurement of peak memory usage and timing without interference from distributed communication overhead. In practice, we used a batch size of 1800 (600 per GPU, trained on 3 GPUs) for ResNet-50 and ViT-base, so we estimate the overhead ratio should be even lower.
>
> As also noted in Sec. 5, reducing the memory footprint of Koopman approximations through streaming DMD is a promising direction for future work.

---

> ### Author Response · Authors · 2025-11-23
> **Response to Reviewer sCUV (Part3)**
>
> **Question 1: In figure 4, is the base optimizer SGD for PDT?**
>
> Yes, in Figure 4, the base optimizer for PDT is SGD. In fact, all the comparisons in the paper are between PDT and its base optimizer.
>
> **Question 2: In line 1041, the citation is missing.**
>
> Thank you for pointing this out! The missing citation should reference "Sec. A.9". We have fixed this in the revised manuscript.

---

### Author Response · Authors · 2025-12-04
**Summary of Rebuttal and Revisions**

We thank all reviewers for their constructive feedback and insightful questions. All these have greatly helped us improve the quality of the work. Here, we summarize the changes made to the paper, highlight the substantial improvements and additional experiments conducted during the rebuttal period. We believe these revisions have addressed all the concerns raised by reviewers.

**1. Clarification of Theoretical Foundations (Reviewer efcL, SwkT, JHov)**
- We added a detailed derivation of the DMD algorithm in Appendix A.2.1.
- We revised Sec. 3.2 to clarify the connection $(W_i, W_{i+1}) \to A \to (\Phi, \Lambda)$ more explicitly.
- We fixed a typo in the acceleration effectiveness criterion (Eq.6).
- We expanded the convergence proof with a complete derivation in Appendix A.5.
- We discussed more work on predicting future weights and their limitations in Sec.2.

**2. Runtime/Memory Overhead & Scalability (Reviewers sCUV, SwkT)**
- We added a memory complexity analysis and a detailed breakdown of the runtime complexity analysis in Appendix A.4.
- We added a detailed "Profiling and Overhead Analysis" section in Appendix A.10. The profiling results align robustly with our theoretical analysis, confirming that PDT is scalable to large modern architectures.

**3. Masking Mechanism Ablation Study (Reviewer SwkT)**
- We added a comprehensive ablation study for masking criterion in Appendix A.11.

**4. Hyperparameter Robustness (Reviewer SwkT, JHov)**
- We added a heuristic guide helping determine the appropriate hyperparameters in Appendix A.9.
- We analyzed the impact of initial learning rates on the masking ratio in Appendix A.12.

**5. Generalization on Non-Vision Tasks (Reviewer JHov)**
- We added an evaluation on natural language processing tasks to validate PDT’s cross-domain generalizability in Appendix A.13.

We have uploaded the revised PDF reflecting all these changes. These additions robustly support the validity and efficiency of our proposed Predictive Differential Training.

---

### Meta-Review · Area_Chair_URcm · 2026-01-06

**Summary:**

The reviewers agree that the paper tackles an interesting and practically relevant problem: accelerating SGD-style training via Koopman/DMD-based predictive differential updates, with an empirically strong plug-in method applicable to many optimizers and architectures. However, several concerns informed my deliberation:

- **Methodological clarity and correctness (efcL, partially SwkT, JHov):** The initial submission did not clearly explain how the Koopman/DMD operator is constructed, how eigen-decompositions are used in practice, and how the masking mechanism is defined and justified. There was confusion around key equations, the role of the snapshot matrices, and the convergence argument.

- **Computational overhead and scalability (sCUV, SwkT, JHov):** Reviewers were worried about runtime and especially memory costs of repeated SVDs and storing multiple weight snapshots, and whether the claimed 10–40% speedups remain meaningful on large-scale models.

- **Design and robustness of the masking strategy and hyperparameters (efcL, SwkT, JHov):** Questions included how often predictions are applied, how sensitive performance is to the prediction interval and window size, and whether both masking criteria (acceleration effectiveness vs dynamic consistency) are truly necessary.

- **Experimental scope and positioning vs prior work (sCUV, JHov):** Concerns included (i) use of non-SOTA training recipes leading to sub-SOTA ImageNet accuracies, and (ii) limited comparison to prior weight-prediction approaches and to non-vision domains.

After rebuttal, I view the technical foundation and empirical support as substantially strengthened. Remaining reservations relate mostly to practicality (notably memory overhead in some regimes) and the complexity of the method and analysis for a general audience, but these do not outweigh the clear novelty and breadth of evidence.

**Reviewer Concerns:**

Addressed or largely addressed:

- **Methodological clarity and DMD details (efcL, SwkT, JHov):** The authors added a detailed derivation of the DMD algorithm and its implementation for weight trajectories (new Appendix A.2.1), explaining how the snapshot matrices, low-rank proxy, eigen-decomposition, and reconstruction of modes interact, and clarifying that the high-dimensional operator is never formed explicitly. They also clarified how projections onto non-orthogonal modes are computed via least-squares and how this leads to the prediction formula, directly answering efcL’s follow-up questions. They expanded the convergence analysis (Appendix A.5) into a full derivation based on L-smoothness assumptions and the two masking criteria, and fixed the inequality mismatch between Eq. 6 and Eq. 8 in the main text. SwkT’s and JHov’s concerns about the proof are thus substantially addressed.

- **Runtime and memory overhead (sCUV, SwkT, JHov):** The rebuttal includes explicit space and time complexity analysis and extensive profiling tables for AlexNet/CIFAR-10, ResNet-50/ImageNet-1k, and ViT-Base/ImageNet-1k (Appendix A.4 and A.10). These show negligible runtime overhead (around 0.01–0.1% per epoch on large-scale models) and moderate memory overhead for modern networks (13–26% extra GPU memory for ResNet-50 and ViT-Base). This clarifies that the method is runtime-efficient and reasonably scalable on large architectures, though memory usage can be substantial for small models.

- **Masking mechanism and hyperparameters (efcL, SwkT, JHov):** The authors provide a precise explanation of why the mask is binary and how the two criteria are complementary for both stability and acceleration. A new ablation study shows that using only one of the criteria leads to severe instability or divergence, while the full mask yields robust gains (Appendix A.11). They also present “rule of thumb” guidelines and a sensitivity study for the four PDT hyperparameters (snapshot count, prediction steps, interval, and start epoch), and analyze how prediction interval and snapshot count affect performance (Appendix A.9). Additional experiments explore how the masked ratio evolves with training and with different learning rates (Appendix A.7, A.12), addressing JHov’s questions about optimizer behavior and LR effects.

- **Experimental scope and prior work (JHov, sCUV):** The revised version expands the related-work section to more thoroughly position PDT relative to prior weight-prediction approaches such as Introspection and NiNo, explaining differences in meta-training requirements and inference overhead. The authors also add cross-domain experiments on an LSTM for AG News text classification, demonstrating consistent speedups in an NLP setting (Appendix A.13). These additions alleviate concerns about narrow applicability. The issue of non-SOTA ImageNet accuracy is addressed by clarifying that the primary goal is relative speedup under fixed, simple recipes and by arguing that PDT is compatible with more advanced training pipelines; while additional SOTA-level runs would be nice, I do not view this as essential for acceptance given the breadth of current experiments.

Remaining or partially outstanding concerns:

- **Memory footprint on smaller models:** For AlexNet/CIFAR-10, the peak memory overhead is more than 10x baseline due to snapshot storage and SVD workspace, even though the overhead ratio shrinks on larger networks. This may limit immediate deployment in low-resource settings, and options such as streaming DMD are deferred to future work.

- **Overall complexity for general readers:** Even after the revisions, understanding the full pipeline (Koopman/DMD derivation, mask design, and convergence proof) requires nontrivial background. This is inherent to the topic but means the paper will be more accessible to readers with some control/DMD familiarity.

On balance, the rebuttal addresses the core scientific concerns to my satisfaction; the remaining issues are more about practicality and presentation than soundness.

**Reviewer Scores:**

**Reviewer efcL:** This reviewer started with a strong reject (rating 2) due to serious confusion about the method and its implementation. After a first round of responses and rereading, they already indicated the idea is interesting and raised the score to 4, then after the detailed DMD derivation and efficiency clarifications explicitly wrote that they were satisfied and “have raised my score to recommend acceptance.” The authors’ summary notes a final move to 6. I agree with this trajectory and expect the final score to be around 6.

**Reviewer sCUV:** sCUV initially gave a 4 (“marginally below acceptance”) mainly because of unclear runtime definition, lack of memory analysis, and concerns about non-SOTA training runs. The rebuttal thoroughly clarifies that runtime numbers include prediction overhead, provides detailed profiling confirming negligible time cost, adds a clear memory complexity treatment, and explains the experimental setup and the rationale for not targeting SOTA recipes. I expect these clarifications would move this reviewer to a positive but not enthusiastic stance, likely raising their score to 6.

**Reviewer SwkT:** SwkT already gave a strong positive score of 8, with concerns limited to the convergence proof, memory complexity, masking ablations, and hyperparameter guidance. All of these points were addressed in depth in the rebuttal and revisions. I expect this reviewer would maintain their score of 8, possibly with even higher confidence.

**Reviewer JHov:** JHov’s main reservations were incomplete related-work coverage, limited discussion of the convergence proof, lack of analysis of masked ratio and learning-rate effects, and missing non-vision experiments. The authors addressed all of these with expanded related work, a more complete convergence derivation, detailed masked-ratio and learning-rate experiments, and new NLP experiments. Given that JHov’s original rating was 4, I believe these substantial additions would shift their view to positive, likely raising the score to 6.

Taking all reviews and the rebuttal into account, I recommend acceptance as a poster. The paper delivers a novel and practically useful predictive-training framework, now supported by clearer theory, detailed implementation description, and extensive empirical evidence across architectures and domains.

---

### Decision · Program_Chairs · 2026-01-26

Accept (Poster)